# Heart of glass anchors Rasip1 at endothelial cell-cell junctions to support vascular integrity

Bart-Jan de Kreuk[1†], Alexandre R Gingras[1†], James DR Knight[2], Jian J Liu[1], Anne-Claude Gingras[2,3]*, Mark H Ginsberg[1]*

[1]Department of Medicine, University of California, San Diego, San Diego, United States; [2]Lunenfeld-Tanenbaum Research Institute, Mount Sinai Hospital, Toronto, Canada; [3]Department of Molecular Genetics, University of Toronto, Toronto, Canada

**Abstract** Heart of Glass (HEG1), a transmembrane receptor, and Rasip1, an endothelial-specific Rap1-binding protein, are both essential for cardiovascular development. Here we performed a proteomic screen for novel HEG1 interactors and report that HEG1 binds directly to Rasip1. Rasip1 localizes to forming endothelial cell (EC) cell-cell junctions and silencing HEG1 prevents this localization. Conversely, mitochondria-targeted HEG1 relocalizes Rasip1 to mitochondria in cells. The Rasip1-binding site in HEG1 contains a 9 residue sequence, deletion of which abrogates HEG1's ability to recruit Rasip1. HEG1 binds to a central region of Rasip1 and deletion of this domain eliminates Rasip1's ability to bind HEG1, to translocate to EC junctions, to inhibit ROCK activity, and to maintain EC junctional integrity. These studies establish that the binding of HEG1 to Rasip1 mediates Rap1-dependent recruitment of Rasip1 to and stabilization of EC cell-cell junctions.

*For correspondence: gingras@
lunenfeld.ca (ACG); mhginsberg@
ucsd.edu (MHG)

[†]These authors contributed
equally to this work

**Competing interests:** The
authors declare that no
competing interests exist.

**Reviewing editor:** Kari Alitalo,
University of Helsinki, Finland

## Introduction

Precisely regulated cell-cell adhesion is crucial for the development and functioning of multicellular organisms (*Nelson, 2008*). In vascular endothelial cells (EC), cell-cell contacts formed by adherens and tight junctions (*Bazzoni, 2004*), control vascular patterning and provide a barrier to the passage of water and solutes. Weakening of this endothelial barrier contributes to diseases such as sepsis, atherosclerosis, and multiple sclerosis (*Bazzoni, 2004*; *Dejana et al., 2009*). The actin cytoskeleton (*Mège et al., 2006*) and its regulatory Rho GTPases have profound roles in the assembly and disassembly of cell-cell contacts (*Beckers et al., 2010*; *Yamada and Nelson, 2007*). Rap1, a small GTPase, regulates epithelial and EC cell-cell adhesion (*Pannekoek et al., 2014*; *Kooistra et al., 2007*; *Knox, 2002*). In EC, binding of activated Rap1 to KRIT1 (*Gingras et al., 2013*; *Li et al., 2012*), a scaffold that regulates RhoA signaling (*Whitehead et al., 2009*; *Stockton et al., 2010*), targets KRIT1 to cell-cell contacts (*Liu et al., 2011*; *Glading et al., 2007*; *Béraud-Dufour et al., 2007*) where it directly interacts with Heart of Glass (HEG1) (*Gingras et al., 2012*), a transmembrane receptor essential for cardiovascular development (*Mably et al., 2003*; *Kleaveland et al., 2009*). The 111 amino acid HEG1 cytoplasmic domain is highly conserved (e.g. 76% identity between human and zebrafish) whereas KRIT1 binding requires only the C-terminal 10% of HEG1 tail (*Gingras et al., 2012*). Thus, we suspected that other interactors of HEG1 might contribute to vascular function. Here, we report an unbiased proteomic screen for HEG1-binding endothelial-specific proteins that revealed that Rasip1, a second Rap1 binding protein essential for vascular integrity and development (*Post et al., 2013*; *Xu et al., 2009*; *Xu et al., 2011*; *Wilson et al., 2013*), directly interacts

**eLife digest** Blood vessels are lined with cells known as vascular endothelial cells. These cells are connected to each other at junctions that consist of several different proteins. The junctions help to control how the blood vessel develops and provide a barrier that controls the movement of water and certain other molecules through the vessel wall. This barrier becomes weakened in diseases like sepsis and atherosclerosis.

Two proteins that are essential for the heart and blood vessels to develop correctly are called "Heart of Glass" (HEG1) and Rasip1. Although a protein has been identified that binds to HEG1 at the cell junctions, this binding only involves a small region of HEG1. This led de Kreuk, Gingras et al. to look for other proteins that interact with HEG1 and that might be important for controlling the development of the blood vessels. This revealed that HEG1 binds directly to Rasip1.

Further experiments revealed that HEG1 is essential for targeting Rasip1 to the junctions between the endothelial cells, and that this helps to stabilize the cell junctions. Mutant forms of Rasip1 that lacked a particular sequence in the middle of the protein were unable to bind to HEG1 and did not localize to the cell junctions. These studies open the door to future work to define how the interaction of Rasip1 and HEG1 is controlled and how Rasip1 stabilizes blood vessels.

with the HEG1 cytoplasmic domain independent of KRIT1. Rasip1 recruitment to cell-cell junctions requires the presence of HEG1 and integrity of its HEG1 binding site. Moreover, targeting of HEG1 tail to mitochondria recruits Rasip1 to these organelles through the direct interaction of the proteins. We designed a structure-based Rasip1 RA-domain mutant (R182E) that failed to bind to Rap1 and did not associate with HEG1 in cells or maintain EC barrier integrity. Finally, the central region of Rasip1 interacts with HEG1 and deletion of this domain eliminates Rasip1's ability to translocate to EC junctions, to inhibit Rho Kinase (ROCK) activity, and to maintain EC junctional integrity. These studies reveal that HEG1 interacts with two different Rap1 effectors important in EC, indicating that this transmembrane receptor serves as a physical nexus for Rap1 signaling in vascular integrity and development.

## Results

### Unbiased proteomic screen identified interaction of Rasip1 with HEG1

To test the idea that interactors besides KRIT1 might be important for HEG1 function, we used wild-type (WT) HEG1 cytoplasmic tail and HEG1 lacking the C-terminal Tyr-Phe (ΔYF) required for KRIT1 binding (*Gingras et al., 2012*) as an affinity matrix (*Figure 1A* and *Figure 1—figure supplement 1A*). To discern potential endothelial-specific interactors, we analyzed the proteins bound to these matrices from lysates of Human Umbilical Vein Endothelial Cells (HUVEC) or HeLa cells by mass spectrometry (MS; Raw data is deposited in the MassIVE repository housed at the Center for Computational Mass Spectrometry at UCSD. See Materials and methods section for more information). This analysis identified a Rap1 effector, Rasip1, among the proteins isolated from endothelial cells but not from HeLa cells (*Figure 1*), a result consistent with the endothelial-specific expression of Rasip1 (*Mitin et al., 2006*; *Mitin et al., 2004*). In contrast KRIT1, and the known KRIT1 interactors CCM2 and ITGB1BP1 (ICAP1) (*Zawistowski, 2005*; *Zawistowski, 2002*) were isolated from both lysates. All of these interactions were specific in that these proteins were detected at 35–125 fold less abundance in a control affinity matrix generated from the αIIb integrin. These proteins were also statistically significant according to SAINTexpress, a method for scoring protein-protein interaction data from affinity purification MS experiments. Moreover, deletion of the last 2 residues of the HEG1 tail abolished binding of KRIT1 as expected, and of the KRIT1-binding proteins CCM2 and ITGB1P1. In sharp contrast, Rasip1 binding was not reduced by the absence of the YF motif (*Figure 1B* and *Figure 1—figure supplement 1B*). Immunoblotting confirmed that endogenous Rasip1 interacts with both the HEG1 cytoplasmic tail and HEG1 ΔYF (*Figure 1C*). To ask whether Rasip1 and HEG1 can interact in living cells, we transfected HUVEC with mCherry-HEG1 cytoplasmic tail, containing a mitochondrial-targeting sequence (*Figure 1D*), and analyzed endogenous Rasip1 distribution.

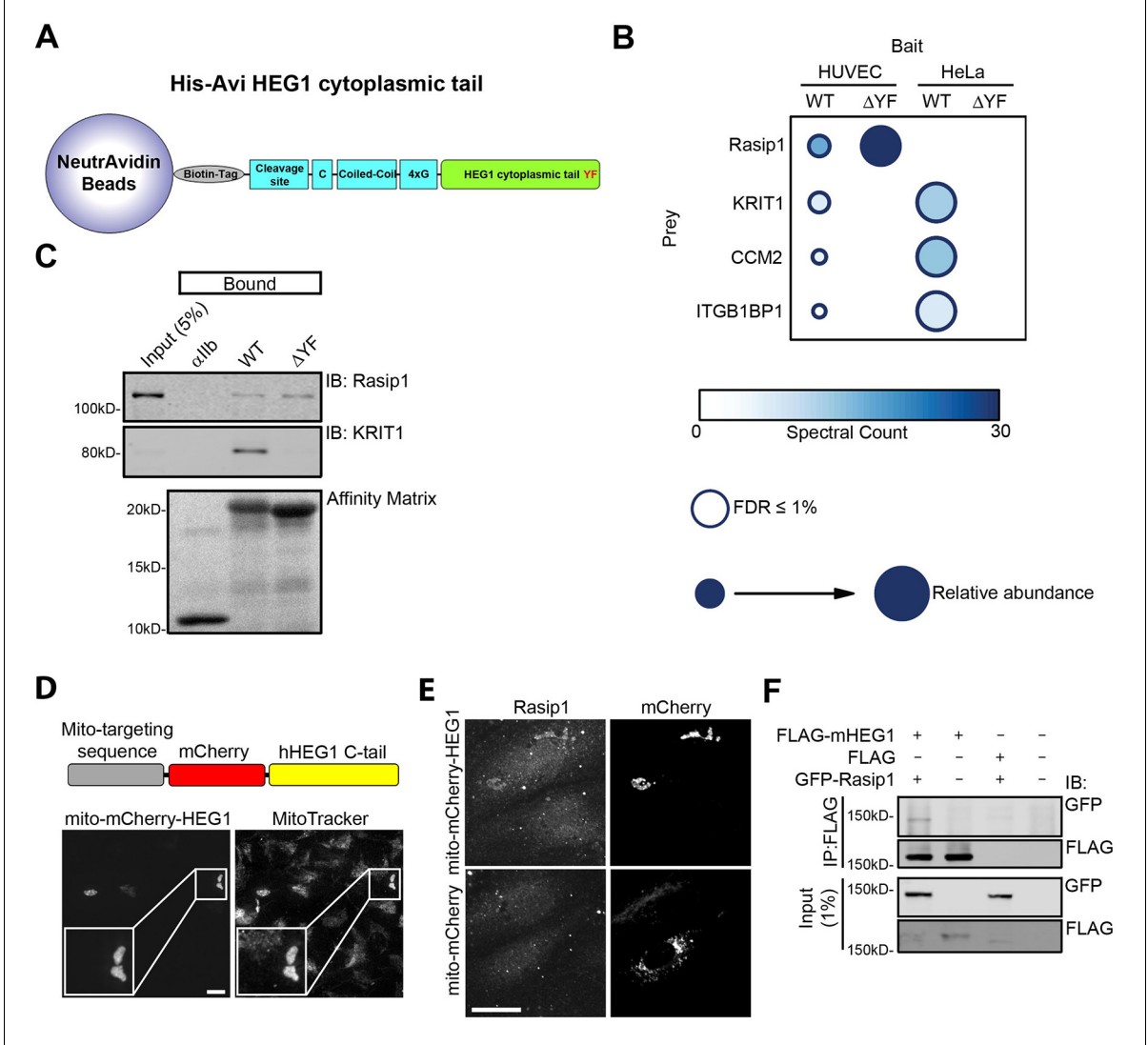

**Figure 1.** Rap1 effector Rasip1 interacts with the transmembrane protein Heart-of-Glass 1. (**A**) Schematic representation of the Heart-of-Glass 1 cytoplasmic tail (aa1274-1381) peptide coupled to NeutrAvidin beads. The C-terminal YF motif is indicated in red. (**B**) Mass spectrometry analysis was performed to identify novel HEG1 interactors. HEG1 wild-type or ΔYF cytoplasmic tail coupled to NeutrAvidin beads was used as bait using lysates from Human Umbilical Vein Endothelial Cells (HUVEC) or HeLa cells. HEG1 wild-type as well as ΔYF bound to Rasip1 from HUVEC lysate but not from HeLa lysate. In contrast, KRIT1, CCM2, and ITGB1BP1 (ICAP1) bound to wild-type HEG1 but not ΔYF in both HUVEC and HeLa. Color and size of dots indicate spectral count and relative abundance respectively (*Antonio Vizcaíno et al., 2015*). False-Discovery-Rate (FDR) was less than 1%. (**C**) Western blot analysis shows that wild-type (WT) HEG1 cytoplasmic tail and ΔYF bound to Rasip1 from HUVEC lysates, whereas KRIT1 only bound to wild-type HEG1 cytoplasmic tail. αIIb cytoplasmic tail was used as a control. Affinity Matrix was visualized by Ponceau staining. Data are representative of at least 3 independent experiments. (**D**) Top section: Schematic representation of HEG1 cytoplasmic tail fused to mCherry fluorescent protein and a mitochondrial targeting sequence. Bottom section: HUVECs, transfected with mito-mCherry-HEG1, were incubated with DeepRed-conjugated Mitotracker (500 nM; 30 minutes) and analyzed by Spinning Disk Confocal Microscopy (SDCM). Mito-mCherry-HEG1 intracellular distribution colocalized with mitochondria as visualized by Mitotracker. Higher magnification images of the boxed area are included. Scale bars, 10 μm. (**E**) HUVECs, transfected with mito-mCherry-HEG1 wild-type (WT) or mito-mCherry alone, were analyzed by SDCM for endogenous Rasip1 localization. A fraction of Rasip1 was targeted to mito-mCherry-HEG1 positive structures but not to mito-mCherry. Representative images of 3 independent experiments are shown. Scale bars, 10 μm. (**F**) HEK293T cells were transfected with GFP-tagged full-length Rasip1, FLAG-tagged full-length murine HEG1/FLAG empty vector, or both. Immunoprecipitation was done by using anti-FLAG G1 resin and bound proteins were separated by SDS-PAGE. Western blot analysis shows that wild-type GFP-tagged Rasip1 was co-immunoprecipitated with full-length FLAG-tagged murine HEG1. See also *Figure 1—figure supplement 1*.

The following figure supplement is available for figure 1:

**Figure supplement 1.** Mass spectrometry identifies Rasip1 as a novel HEG1 interactor.

Targeting of mito-mCherry-HEG1, but not mito-mCherry alone, to mitochondria recruited Rasip1, demonstrating the interaction of HEG1 and Rasip1 in cells (*Figure 1E*). In addition, Rasip1 co-immunoprecipitated with full-length HEG1 (*Figure 1F*). Consistent with the mass spectrometry results, immunoblotting confirmed that Rasip1 bound to HEG1 ΔYF (*Figure 1B,C*), suggesting that the HEG1-Rasip1 interaction does not require KRIT1 binding to HEG1. To further test the KRIT1-dependence of the HEG1-Rasip interaction, we silenced KRIT1 expression (*Figure 2A,B*; 70% reduction in expression was achieved) and found that it did not affect binding of GFP-Rasip1 to the HEG1 cytoplasmic tail affinity matrix (*Figure 2A*) or Rasip1 recruitment to mito-mCherry-HEG1 (*Figure 2C*). Conversely, Rasip1 did not interact with the C-terminal 19 residues of HEG1, which are sufficient to bind KRIT1 (*Figure 1—figure supplement 1C*). Thus, we report that Rasip1 interacts with HEG1 independently of KRIT1.

## HEG1 is required for localization of Rasip1 to cell-cell junctions

Rasip1 redistributes from the endothelial cell cytoplasm to cell-cell junctions during the process of junction assembly (*Wilson et al., 2013*). Because HEG1 is found in these junctions (*Kleaveland et al., 2009*), we hypothesized that the HEG1-Rasip1 interaction might mediate Rasip1 recruitment. To test this idea, we induced junctional assembly by disrupting adherens junctions with EGTA, followed by recalcification in the presence of the Epac1-specific cAMP analog 8-pCPT-2'-O-Me-cAMP (hereafter called '007') to activate Rap1 (*Enserink et al., 2002*). As expected, during junction assembly, Rasip1 was recruited to VE-cadherin-positive cell-cell contacts (*Figure 3A,B*). In sharp contrast, when HEG1 was silenced (*Figure 3C*) Rasip1 recruitment to cell-cell contacts was prevented (*Figure 3A,B*). To exclude off-target effects, silencing HEG1 expression with a second shRNA (*Figure 3—figure supplement 1A*) or an siRNA specific for HEG1 (*Figure 3—figure supplement 1B*) also prevented Rasip1 recruitment to EC contacts. Previous studies have shown that loss of HEG1-KRIT1 interaction increases Rho-Kinase (ROCK) activity (*Gingras et al., 2012*), however, a ROCK inhibitor, H-1152, did not restore Rasip1 recruitment in HEG1-silenced EC (*Figure 3—figure supplement 2A,B*). Moreover, silencing of KRIT1 (*Figure 2B*) did not prevent Rasip1 recruitment to cell-cell contacts during junction formation (*Figure 3D*). Thus, HEG1 mediates Rasip1 recruitment to forming EC junctions.

## HEG1 does not regulate Rasip1-Rap1 or Rasip1-Radil-ARHGAP29 complex formation in vitro

At cell-cell contacts, Rasip1, in conjunction with the Rap1 effector Radil, suppresses RhoA/ROCK signaling via the RhoA RhoGAP ARHGAP29 (*Post et al., 2013*; *Xu et al., 2011*; *Post et al., 2015*). As HEG1 interacts with Rasip1, we tested whether HEG1 may also be involved in regulating Rasip1-Radil-ARHGAP29 complex formation. As expected, Radil and ARHGAP29 co-immunoprecipitated with Rasip1 (*Figure 3—figure supplement 3A,B*). Silencing HEG1 expression (*Figure 3—figure supplement 3C*; 75% reduction in mRNA expression was achieved) did not affect the interaction between Rasip1 and Radil or ARHGAP29 (*Figure 3—figure supplement 3A, B*). Next we tested whether HEG1 could regulate the interaction between Rasip1 and Rap1. Rasip1 interacted with constitutively active Rap1-V12; the interaction was not affected by silencing HEG1 expression (*Figure 3—figure supplement 3C,D*). Similarly, addition of purified HEG1 cytoplasmic tail peptide (5 μM) did not alter the interaction between Rap1-V12 and Rasip1 (*Figure 3—figure supplement 3D*). These data suggest that HEG1 is required for Rasip1 translocation to cell-cell contacts; however, formation of a Rasip1-Rap1 or Rasip1-Radil-ARHGAP29 complex is not dependent on the presence of HEG1.

## Defining the Rasip1-HEG1 binding interface

To analyze the functional importance of Rasip1 binding to HEG1, we mapped the HEG1 region that binds to Rasip1 using truncations and deletions of the HEG1 cytoplasmic tail (*Figure 4A–C*). Deletion of 9 amino acids in HEG1 (aa 1327-1335; TDVYYSPTS) prevented binding of Rasip1 to HEG1 *in vitro* (*Figure 4C*), Rasip1 recruitment to mito-mCherry-HEG1 (*Figure 4D*), and co-immunoprecipitation of Rasip1 with murine HEG1(Δ1283-1291) (corresponding to aa 1327-1335 in human HEG1; *Figure 4E*). Furthermore, a 22-mer encompassing this region (HEG1 1318-1339), which lacked KRIT1 binding, was sufficient to interact with Rasip1 (*Figure 4B,C*). Thus, we mapped the Rasip1 binding

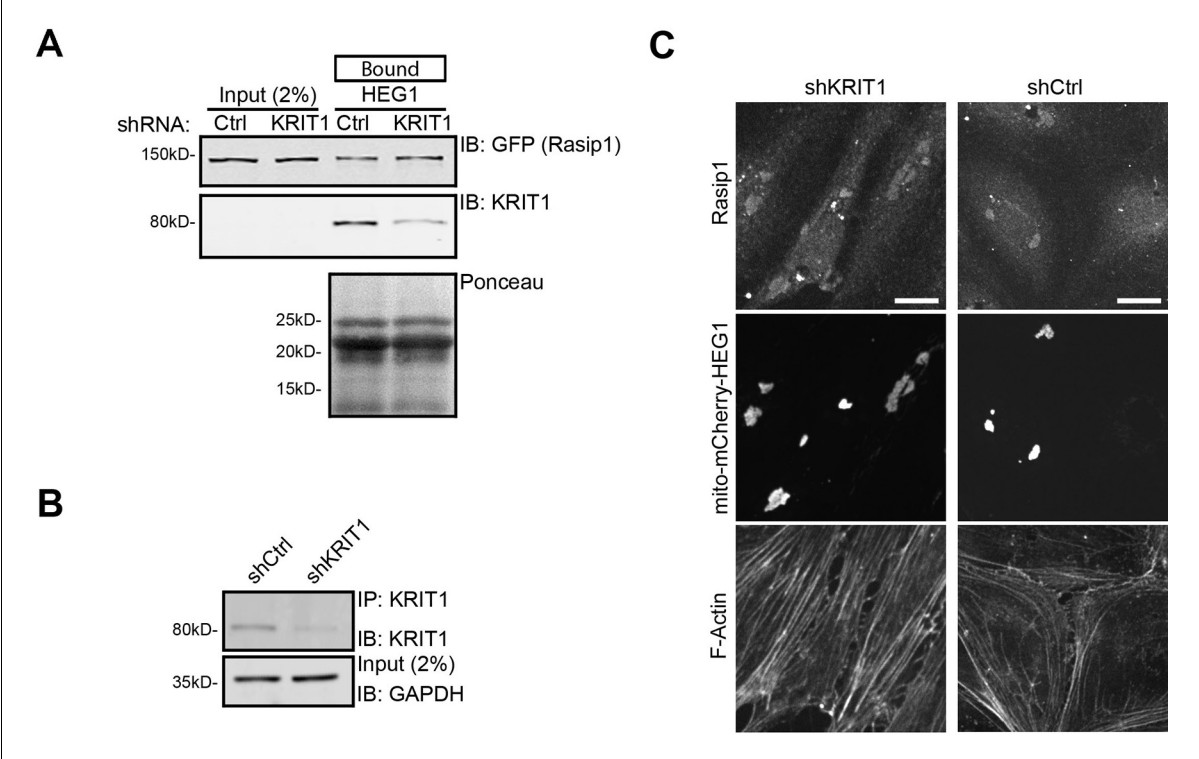

**Figure 2.** Rasip1 localization and binding to HEG1 is independent of KRIT1. (**A**) KRIT1 expression in HEK293T was reduced by lentiviral expression of shKRIT1. Western blot analysis shows that wild-type HEG1 cytoplasmic tail binds to GFP-tagged Rasip1 from HEK293T lysates independent of KRIT1 protein expression. Affinity Matrix was visualized by Ponceau staining. Data are representative of 3 independent experiments. (**B**) KRIT1 Immuno-Precipitation (IP) was performed to visualize KRIT1 protein expression in Human Umbilical Vein Endothelial Cells (HUVEC). Western blot analysis shows that KRIT protein expression is reduced in shKRIT1-expressing cells. GAPDH was used a loading control. (**C**) HUVECs, infected with control shRNA (shCtrl) or KRIT1-specific shRNA (shKRIT1) and subsequently transfected with mito-mCherry-HEG1, were analyzed by Spinning Disk Confocal Microscopy (SDCM) for endogenous Rasip1 localization. Independent of KRIT1 protein levels , a fraction of wild-type Rasip1 was targeted to mito-mCherry-HEG1 positive structures. See Panel B for Western blot analysis of KRIT1 protein expression. Higher magnification images of the boxed area are included. Representative images of 3 independent experiments are shown. Scale bars, 10 μm.

site in HEG1 to a 9 residue peptide and show that deletion of this sequence blocks the capacity of HEG1 to bind to and to recruit Rasip1.

Full-length Rasip1 contains an N-terminal poly-Proline region and Ras Association (RA) domain, a central Forkhead-associated (FHA) domain, and a C-terminal Dilute (DIL) domain (*Figure 5A*). We transfected HEK293T cells with FLAG-tagged Rasip1(1-265; poly-Pro+RA), (266-550; FHA), or (551-963; DIL), and measured binding to HEG1 tail affinity matrix. Rasip1(266-550), containing the FHA domain, was sufficient for binding to HEG1 (*Figure 5B*). Next, we tested whether this region is necessary for binding to HEG1. Deletion of this region in Rasip1(Δ334-539) (*Figure 5A*) disrupted HEG1 binding (*Figure 5C*). Thus, the region of Rasip1 encompassing the FHA domain is both necessary and sufficient to bind to HEG1. Furthermore, the interaction of HEG1 and the FHA domain was direct because purified Rasip1(266-550) bound to both HEG1, HEG1 ΔYF, or HEG1(1318-1339) peptide affinity matrices (*Figure 5D*). Thus, HEG1 binds directly to the FHA domain of Rasip1 via a 9 amino acid (TDVYYSPTS) region of HEG1.

## The direct interaction between Rasip1 and HEG1 is important for Rasip1 junctional localization, regulation of ROCK, and vascular integrity

We studied recruitment of Rasip1 to assembling cell-cell junctions in HUVEC to test the functional relevance of the HEG1-Rasip1 interaction. As described above, during junction assembly, Rasip1 was recruited to endothelial cell-cell contacts. Deletion of the HEG1-binding central domain of Rasip1

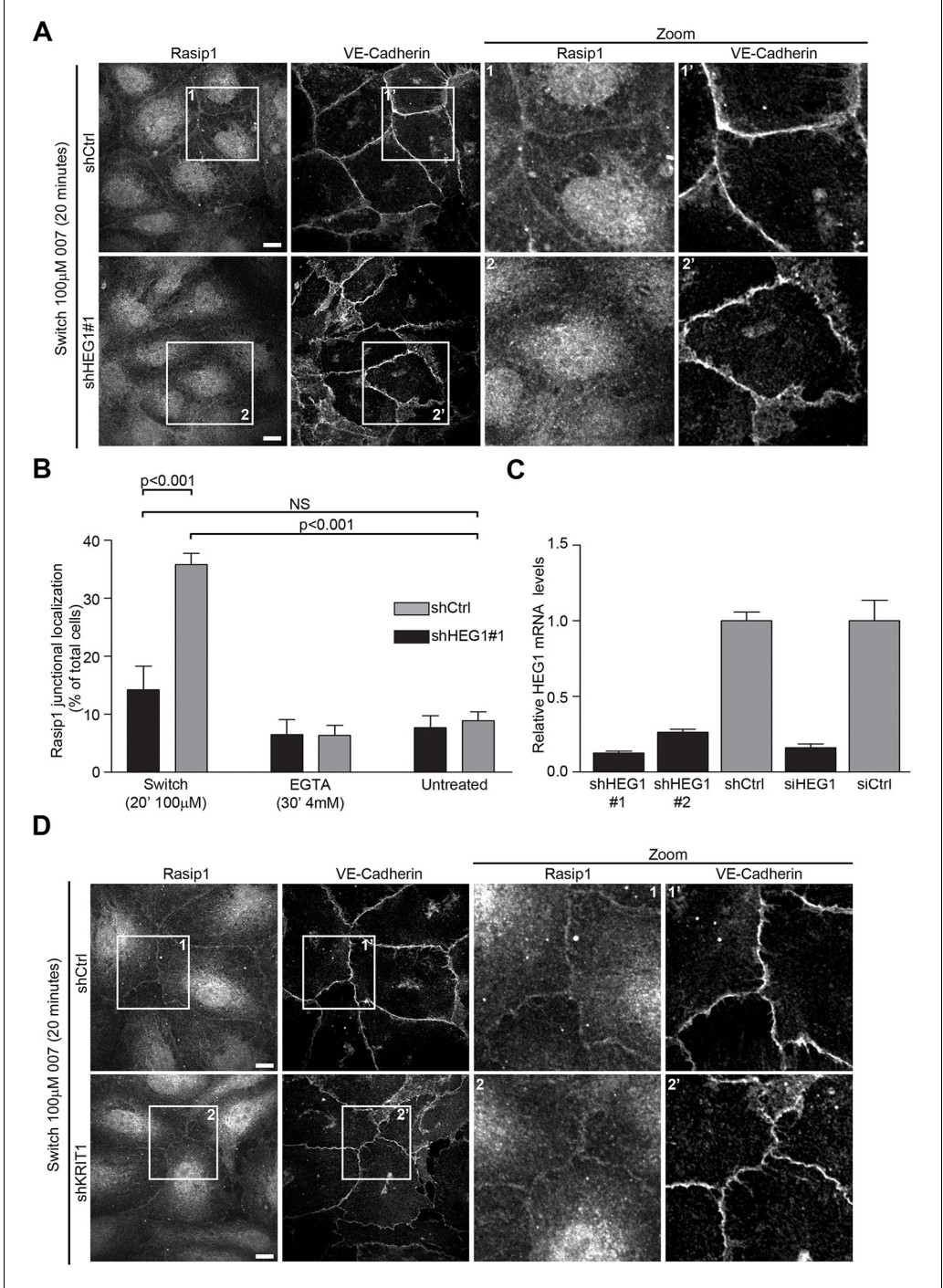

**Figure 3.** Rasip1 junctional localization is dependent on HEG1. (**A**) Rasip1 intracellular distribution was analyzed by Spinning Disk Confocal Microscopy (SDCM) in Human Umbilical Vein Endothelial Cells (HUVEC) infected with lentiviral particles containing control shRNA (shCtrl) or HEG1-specific shRNA (shHEG1#1). Cells were treated with DMEM (5% FBS, 4 mM EGTA) to remove Calcium and disrupt adherens junctions. Subsequently, cells were incubated with DMEM containing 8-pCPT-2-O-Me-cAMP-AM ('007', 100 µM) and Calcium (2 mM) for 20 minutes to mimic junction formation/ stabilization. Under these conditions, endogenous Rasip1 localized to cell-cell contacts in control cells (shCtrl). In contrast, shRNA-mediated depletion of HEG1 (shHEG1#1) prevented Rasip1 junctional localization. Higher magnification images of the boxed area are included. Representative images of 3 independent experiments are shown. Scale bars, 10 µm. (**B**) Bar diagram shows percentage of cells with Rasip1 junctional localization. Mean values ± SEM are shown. One-way analysis of variance (ANOVA) with Bonferroni's test was used to compare each condition versus untreated control (shCtrl) cells. Data are from 3 independent experiments. (**C**) Efficiency of HEG1 mRNA depletion in HUVEC was measured by Q-PCR for cells infected with lentiviral particles containing control shRNA or two different shRNAs specific for HEG1 (shHEG1#1 and shHEG1#2) or transfected with control (siCtrl) or

*Figure 3 continued on next page*

*Figure 3 continued*

HEG1-specific siRNA (siHEG1). Mean values ± SEM are shown from 3 independent measurements. (D) Rasip1 intracellular distribution was analyzed by Spinning Disk Confocal Microscopy (SDCM) in HUVEC infected with lentiviral particles containing control shRNA (shCtrl) or KRIT1-specific shRNA (shKRIT1). Cells were treated with DMEM (5% FBS, 4 mM EGTA) to remove Calcium and disrupt adherens junctions. Subsequently, cells were incubated with DMEM containing 8-pCPT-2-O-Me-cAMP-AM ('007', 100 μM) and Calcium (2 mM) for 20 minutes to mimic junction formation/stabilization. Under these conditions, endogenous Rasip1 localization to cell-cell contacts was observed in both control cells (shCtrl) and in KRIT1-depleted cells (shKRIT1). See Figure 2 Panel B for Western blot analysis of KRIT1 protein expression. Higher magnification images of the boxed area are included. Representative images of 3 independent experiments are shown. Scale bars, 10 μm. See also *Figure 3—figure supplements 1–3*.

The following figure supplements are available for figure 3:

**Figure supplement 1.** Rasip1 junctional localization is dependent on HEG1.

**Figure supplement 2.** ROCK inhibition does not restore Rasip1 junctional localization in HEG1 depleted cells.

**Figure supplement 3.** Knock-down of HEG1 does not affect Rasip1-Rap1 or Rasip1-Radil-ARHGAP29 complex formation.

abolished recruitment of Rasip1(Δ334-539) to these junctions (*Figure 5E*). Rasip1 is known to associate with cytoplasmic vesicles (*Xu et al., 2011*; *Mitin et al., 2004*). Remarkably, Rasip1(Δ334-539) accumulated on cytoplasmic vesicles, which often appeared to concentrate near cell-cell junctions; however, in contrast to the full length protein, Rasip1(Δ334-539) never incorporated into the junctions. Thus, the region of Rasip1 that mediates its physical interaction with HEG1 is required for Rasip1 junctional localization.

Rasip1 regulates β1 integrin activation (*Xu et al., 2011*). To investigate whether HEG1 is involved in regulating β1 integrin activation as well, we silenced HEG1 or Rasip1 expression in HUVEC and measured the binding of 9EG7 monoclonal antibody as a reporter of β1 integrin activation (*Lenter et al., 1993*). As expected, silencing Rasip1 expression in HUVEC decreased 9EG7 binding. In contrast, silencing HEG1 expression did not affect levels of activated β1 integrin (*Figure 6A*) suggesting that the effect of Rasip1 on β1 integrin activation is independent of HEG1. Previous studies show that Rasip1 mediates Rap1 inhibition of RhoA activity and of the RhoA effector Rho kinase (ROCK). As a result, silencing Rasip1 expression in endothelial cells can increase phosphorylation of a ROCK substrate, myosin light chain 2 (MLC) (*Xu et al., 2011*), resulting in increased actin stress fibers and reduced cortical actin (*Post et al., 2013*). To test whether HEG1 is also involved in suppressing ROCK signaling we silenced HEG1 or Rasip1 expression in HUVEC and analyzed MLC phosphorylation. Similar to silencing Rasip1, silencing HEG1 expression in HUVEC by siRNA or shRNA increased phosphorylation of MLC and formation of stress fibers indicating increased ROCK signaling (*Figure 6B,C*). As both HEG1 and Rasip1 silencing resulted in increased MLC phosphorylation, we tested whether the interaction between HEG1 and Rasip1 is important for the regulation of RhoA/ROCK signaling. Silencing HEG1 expression in HUVEC increased MLC phosphorylation and formation of actin stress fibers, which was rescued by expressing shRNA-resistant murine HEG1 (*Figure 6D*). However, expression of Rasip1-binding deficient murine HEG1(Δ1283-1291) (corresponding to aa 1327-1335 in human HEG1) failed to rescue the increase in MLC phosphorylation and actin stress fiber formation (*Figure 6D*). Flow cytometry analysis showed that both wild-type mHEG1 and mHEG1(Δ1283-1291) were equally expressed in HUVEC (*Figure 6—figure supplement 1*). Conversely, silencing expression of Rasip1 increased MLC phosphorylation by ~40% (*Figure 7A, B*). Expression of shRNA-resistant wild-type Rasip1 (*Figure 7—figure supplement 1*) rescued increased MLC phosphorylation (*Figure 7A,B*). In contrast, HEG1-binding deficient Rasip1(Δ334-539) failed to rescue increased MLC phosphorylation (*Figure 7A,B*) Moreover, expression of Rasip1 (Δ334-539) induced increased MLC phosphorylation even in the absence of Rasip1 silencing, indicating that it could act in a dominant negative manner (*Figure 7C*). These data show that the interaction between Rasip1 and HEG1 is important for regulation of RhoA/ROCK signaling and the actin cytoskeleton in endothelial cells.

Silencing of Rasip1 increases endothelial permeability (*Post et al., 2013*; *Post et al., 2015*). As expected, silencing Rasip1 increased endothelial permeability, and this was rescued by expression of shRNA-resistant wild-type (WT) Rasip1 (*Figure 7D,E*). In contrast, HEG1-binding deficient Rasip1

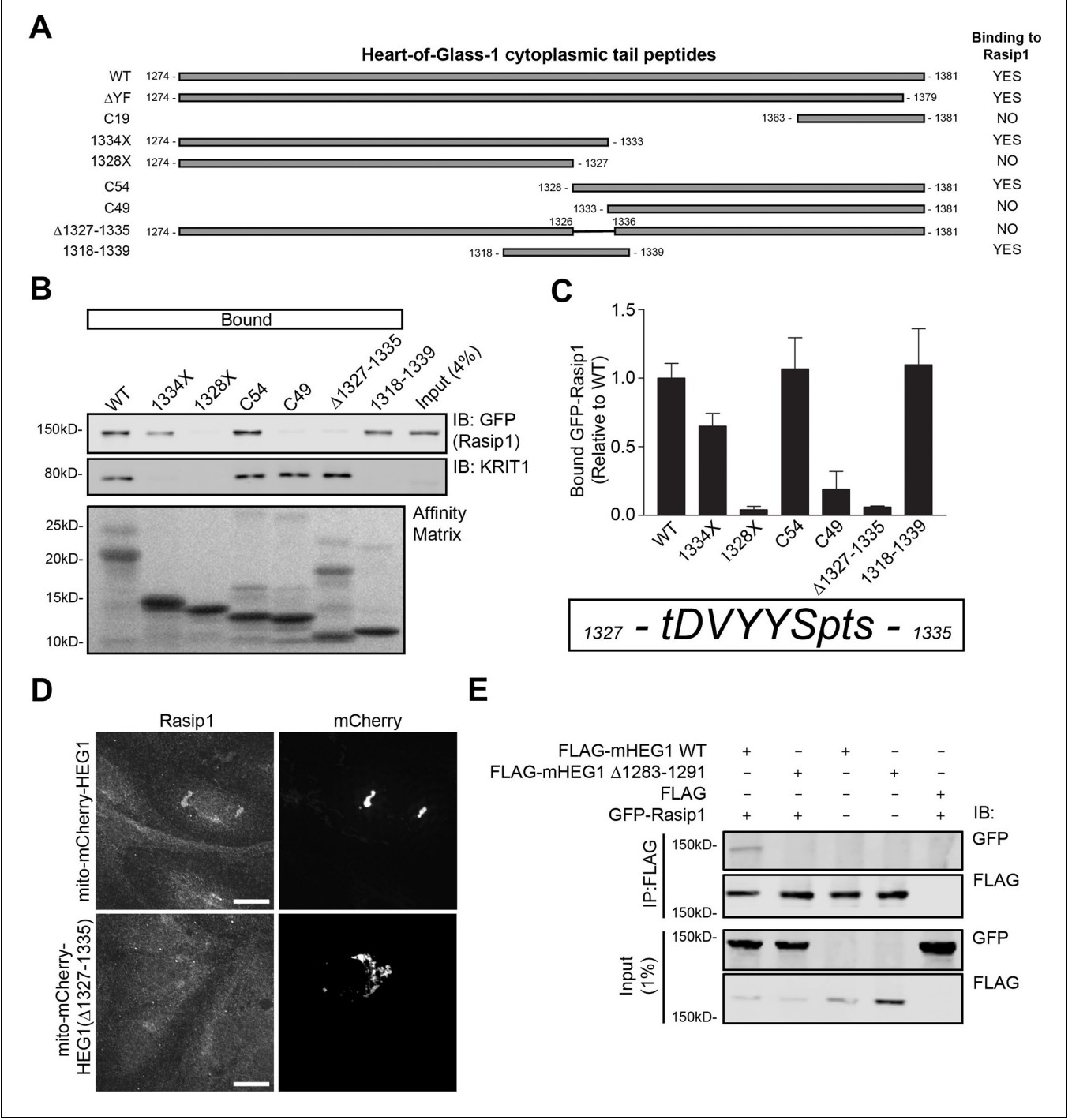

**Figure 4.** Rasip1 binds to HEG1 upstream of the KRIT1-binding site. (**A**) Schematic representation of different HEG1 cytoplasmic tail peptides used to map the binding region for Rasip1. (**B**) HEK293T cells were transfected with GFP-tagged full-length Rasip1. Western blot analysis shows that HEG1 wild-type (WT), 1334X, C54, and 1318-1339 bound to GFP-Rasip1. In contrast, HEG1 1328X, C49, and Δ1327-1335 failed to bind to GFP-Rasip1. Endogenous KRIT1 binding was only observed for HEG1 WT, C54, C49, and Δ1327-1335 which al contain the C-terminal YF motif. Affinity Matrix was visualized by Ponceau staining. (**C**) Top section: Bar diagram shows binding of GFP-Rasip1 to HEG1 cytoplasmic tail peptides relative to wild-type HEG1. Mean values ± SEM are shown from at least 3 independent experiments. Bottom section: HEG1 1327-1335 (TDVYYSPTS) is necessary for Rasip1 binding. (**D**) HUVECs, transfected with mito-mCherry-HEG1 or mito-mCherry-HEG1(Δ1327-1335), were analyzed by Spinning Disk Confocal Miroscopy (SDCM) for endogenous Rasip1 localization. A fraction of Rasip1 was targeted to mito-mCherry-HEG1 positive structures but not to mito-mCherry-HEG1(Δ1327-1335). Scale bars, 10 μm. (**E**) HEK293T cells were transfected with GFP-tagged full-length Rasip1, FLAG-tagged murine HEG1 full-length, Δ1283-1291 (corresponding to aa 1327-–1335 in human HEG1), empty vector, or both. Immunoprecipitation was done by using anti-FLAG G1 resin and bound

*Figure 4 continued on next page*

Figure 4 continued

proteins were separated by SDS-PAGE. Western blot analysis shows that GFP-tagged Rasip1 was co-immunoprecipitated with full-length mHEG1 but not mHEG1(Δ1283--1291).

(Δ334-539) failed to reverse increased endothelial cell permeability (*Figure 7D,E*). Thus, deletion of the HEG1 binding domain of Rasip1 abolishes the capacity of Rasip1 to translocate to cell-cell contacts, to inhibit ROCK, and to maintain EC junctional integrity.

## Rasip1 function is dependent on Rap1-binding

As noted above, Rasip1 binds Rap1 and activation of Rap1 facilitates Rasip1 targeting to cell junctions and suppresses ROCK activity. To test whether Rap1 binding to Rasip1 is required for Rasip1 function and to assess the role of Rap1 in regulating the interaction of Rasip1 with HEG1, we analyzed the Rasip1 Ras-association (RA) domain. Comparison of the Rasip1 RA domain sequence showed that it was similar to those found in RalGDS, Afadin6, and Radil. Sequence alignment of Rasip1 RA domain with Radil shows a highly conserved sequence (31% identical, *Figure 8A*). A preliminary homology model of the Rasip1 RA domain closely resembles the structure of Radil RA domain (PDB ID 3EC8) (backbone rmsd 1.02Å for 126 a.a., *Figure 8B*). The Radil crystal structure shows a dimer that is stabilized by an extensive interface burying 30% (2,324 Å$^2$) of the total surface area of each monomer (*Figure 8B*, highlighted in blue). Analysis of our model shows that the dimer interface is conserved in Rasip1 (*Figure 8B*, magenta and yellow), suggesting that it could also form a dimer similar to Radil RA domain and therefore could bind two Rap1 molecules per dimer (*Figure 8C*). Indeed, purified recombinant Rasip1 RA domain (residues 134-285) eluted earlier than expected from its monomer mass (16.3kDa) in size exclusion chromatography (SEC), suggesting that it forms a dimer (~35kDa) in solution (*Figure 8D*; green). The purified Rasip1 RA domain bound activated Rap1B (*Figure 8D*; red) and Isothermal Titration Calorimetry (ITC) measured a $k_d = 0.77 \pm 0.09$ μM for the interaction (*Figure 8E*).

On the basis of the homology model, we created a Rasip1 RA domain mutant (R182E) mutant that could reduce Rap1 binding and tested its interaction with Rap1. SEC revealed that Rasip1 (R182E) exhibited a markedly reduced affinity for Rap1 (*Figure 8D*; black). Moreover, SEC showed that the Rasip1(R182E) mutant was well folded and exhibited the same elution pattern in SEC as the wild-type protein. Thus, we conclude that Rasip1 RA domain (R182E) remains well folded as a multimer but binds Rap1 with drastically reduced affinity. We examined the capacity of Rasip1(R182E) to localize with HEG1 in cells and found that this mutant was not targeted to mito-mCherry-HEG1 (*Figure 8F*). Thus, in living cells, Rasip1 recruitment to HEG1 is lost when a point mutant disrupts its capacity to bind Rap1. Rasip1(R182E) also failed to rescue the effect of silencing of Rasip1 on increased MLC phosphorylation or endothelial permeability (*Figure 9A–C*) although it was well expressed (*Figure 9D*). Thus, Rasip1(R182E) is well expressed and folded, binds to activated Rap1 with much reduced affinity and loses the capacity to interact with HEG1 in living cells, and to inhibit ROCK and maintain EC junctional integrity.

## Discussion

Heart of Glass (HEG1), a transmembrane protein, is essential for vertebrate cardiovascular development (*Mably et al., 2003*; *Kleaveland et al., 2009*) and localizes KRIT1 to EC junctions. Here, we report that Rasip1, a Rap1 binding protein required for vascular development, binds directly to the cytoplasmic domain (tail) of HEG1, independent of KRIT1. Silencing HEG1 prevented Rasip1 targeting to EC junctions, furthermore the HEG cytoplasmic domain could re-localize Rasip1 in living cells. We mapped the Rasip1 binding site in HEG1 to a 9 residue peptide that is highly conserved in vertebrates and found that deletion of this sequence blocks the capacity of HEG1 to bind to Rasip1 and to recruit Rasip1 in cells. Similarly, we mapped the HEG1 binding site in Rasip1 to its central region, containing an FHA domain, and found that deletion of this domain abolished the capacity of Rasip1 to translocate to cell-cell contacts, to inhibit ROCK activity, and to maintain EC junctional integrity. Rap1 activity regulates the function and localization of Rasip1; using molecular modeling we designed and validated a well-folded mutant, Rasip1(R182E), that failed to bind Rap1. Rasip1

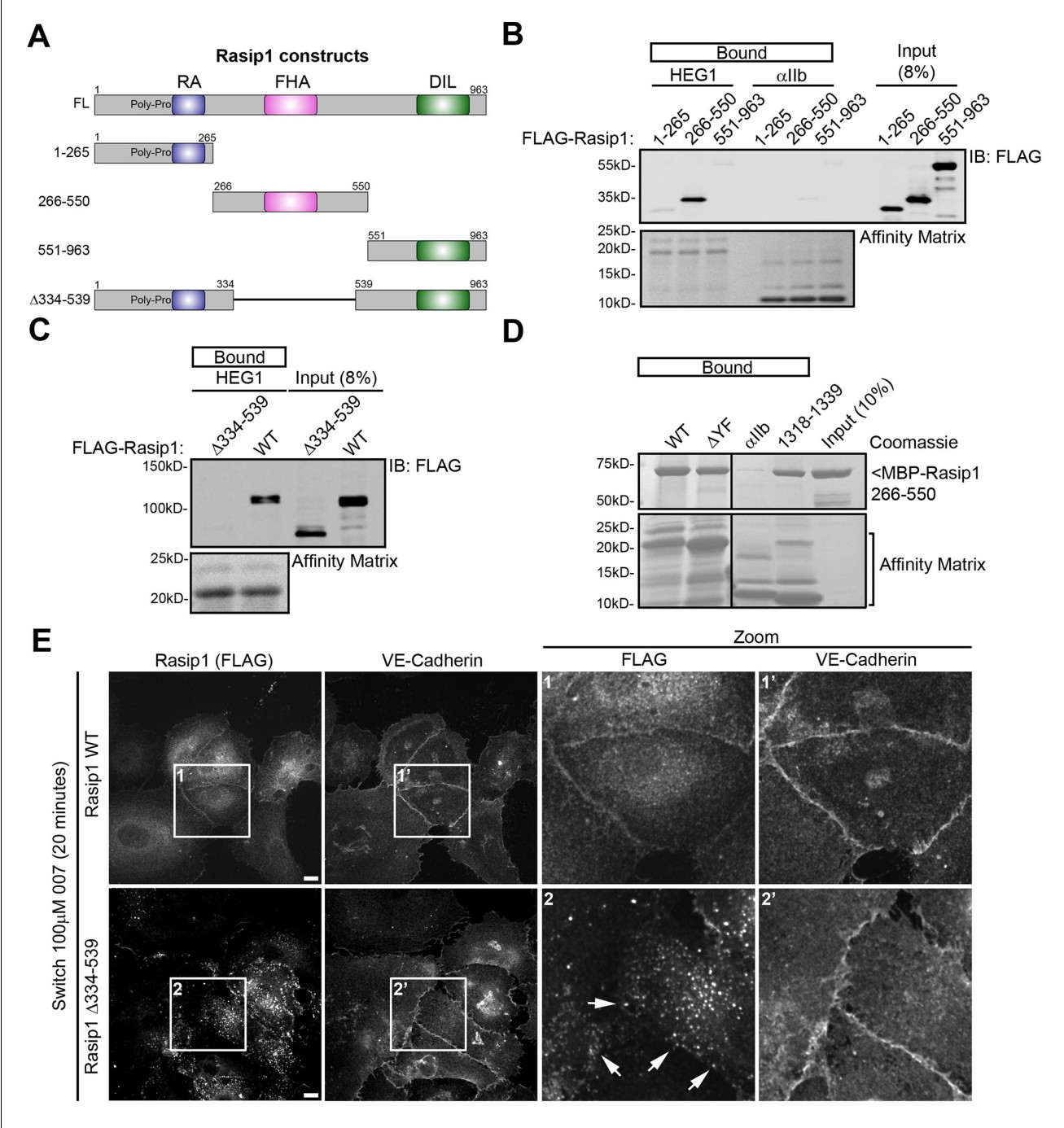

**Figure 5.** Rasip1 central domain interacts with HEG1 cytoplasmic tail. (**A**) Schematic representation of Rasip1 constructs. (**B**) HEK293T cells were transfected with FLAG-tagged Rasip1 1-265, 266-550, or 551-963. Western blot analysis shows that the HEG1 cytoplasmic tail peptide preferentially bound to FLAG-Rasip1 266-550, which contains an FHA domain. αIIb cytoplasmic tail was used as a control. Affinity Matrix was visualized by Ponceau staining. Data are representative of at least 3 independent experiments. (**C**) HEK293T cells were transfected with FLAG-tagged wild-type Rasip1 (WT) or Rasip1(Δ334-539), which lacks the FHA domain. Western blot analysis shows that, in contrast to Rasip1 wild-type, the HEG1 cytoplasmic tail did not interact with Rasip1(Δ334-539). Affinity Matrix was visualized by Ponceau staining. Data are representative of at least 3 independent experiments. (**D**) Wild-type (WT) HEG1 cytoplasmic tail peptide, ΔYF, and HEG1 1318-1339, but not αIIb cytoplasmic tail, directly bound to recombinant MBP-Rasip1 266-550 fusion protein. Coomassie blue-stained SDS-PAGE gel is representative of 3 independent experiments. All lanes were from the same gel. (**E**) FLAG-Rasip1 intracellular distribution was analyzed by Spinning Disk Confocal Microscopy (SDCM) in Human Umbilical Vein Endothelial Cells (HUVEC) expressing FLAG-tagged wild-type (WT) Rasip1 or Rasip1(Δ334-539) expressed by lentiviral infection. Cells were treated with DMEM (5% FBS, 4 mM EGTA) to remove Calcium and disrupt adherens junctions. Subsequently, cells were incubated with DMEM containing 8-pCPT-2-O-Me-cAMP-AM ('007',

*Figure 5 continued on next page*

*Figure 5 continued*

100 μM) and Calcium (2 mM) for 20 minutes to mimic junction formation/stabilization. Under these conditions, Rasip1 WT localized to cell-cell contacts in. In contrast, Rasip1(Δ334-539) failed to localize to junctions albeit Rasip1-positive vesicular structures could be found in the vicinity of cell-cell contacts (indicated by the arrows). Higher magnification images of the boxed area are included. Representative images of 3 independent experiments are shown. Scale bars, 10 μm.

(R182E) also failed to associate with the HEG1 tail in cells and lost the capacity to inhibit ROCK and maintain EC junctional integrity. These studies establish that the interaction of HEG1 and Rasip1, two proteins involved in vascular development, is regulated by Rap1 binding to Rasip1 and enables Rasip1 to localize to EC cell-cell junctions and to maintain vascular integrity.

The binding of a Rasip1 FHA domain to a 9 amino acid (TDVYYSPTS) region in HEG1 mediates Rasip1 localization and function in EC. Rasip1 is recruited to assembling EC cell-cell junctions and we observed that HEG1 silencing blocked this recruitment and that targeting mito-mCherry-HEG1 to mitochondria re-localized endogenous Rasip1 to these organelles. Mapping studies identified HEG1(1327-1335) as the critical region and deleting this site abrogated the capacity of HEG1 to bind Rasip1 and to recruit it to mito-mCherry-HEG1(1327-1335). Rasip(266-550), which contains an FHA domain, was necessary and sufficient for the interaction with the HEG1 tail. In sum, our studies show that the direct physical interaction of a HEG1 with Rasip1 targets Rasip1 to assembling EC junctions.

Rasip1 regulation of ROCK activity and vascular permeability is dependent on the interaction with HEG1. In contrast to wild-type Rasip1, HEG1-binding deficient Rasip1(Δ334-539) was not recruited to endothelial cell-cell contacts during junction formation/remodeling. Groundbreaking studies (*Xu et al., 2011*) revealed that Rasip1 interacted with a RhoGAP (ARHGAP29) regulating RhoA and its effector ROCK1, the latter assayed by increased phospho-myosin light chain. More recent studies established that the Rasip1-ARHGAP29 axis also regulates endothelial permeability (*Post et al., 2013*; *Wilson et al., 2013*) in cooperation with Radil, which binds directly to the RhoGAP (*Post et al., 2015*). Our finding that HEG1-binding defective Rasip1(Δ334-539), which did not reach EC junctions, failed to rescue both increased phospho-myosin and vascular permeability caused by Rasip1 silencing, supports the idea (*Post et al., 2015*) that the junctional localization of Rasip1 is important for regulation of RhoA/ROCK and vascular permeability. We noted (*Figure 5E*) that Rasip1(Δ334-539) appears to accumulate on vesicles, suggesting that Rasip1(Δ334-539) transport to the cell periphery is intact but that incorporation into cell-cell contacts depends on HEG1. In sum, our data show that Rasip1-HEG1 interaction is important for Rasip1 junctional localization and Rasip1-mediated regulation of ROCK signaling and endothelial barrier integrity.

The studies reported here establish HEG1 as a physical nexus for Rap1 signaling in the vasculature. We found that Rasip1 and KRIT1 bound to HEG1 at distinct binding sites. Rap1 binding regulates the capacity of both proteins to target to HEG1-containing junctions, inhibit ROCK activity, and maintain junctional integrity (*Figure 10*). Importantly, the *Rasip1* murine null phenotype is lethal by day E9.5 with much more profound vascular collapse (*Xu et al., 2011*; *Wilson et al., 2013*) than is seen in the *Heg1* null (*Kleaveland et al., 2009*) suggesting that Rasip1 has functions that do not depend on HEG1 binding or recruitment to EC junctions. One compelling possibility is that Rasip1 null mice exhibit profound detachment of EC from their basement membranes (*Xu et al., 2011*), perhaps due to inhibition of integrin activation. Indeed we observed that silencing HEG1 did not impair β1 integrin activation, whereas silencing Rasip1 did so. Thus, the Rap1-regulated interaction of HEG1 with Rasip1 localizes Rasip1 to those junctions and stabilizes them. Remarkably, Rap1 regulated association of KRIT1 with a second site in HEG1 also contributes to vascular development and integrity. Thus, HEG1 serves as physical nexus of Rap effectors that control both RhoA/ROCK signaling and vascular development.

## Materials and methods

### cDNA and RNAi

Cloning of point mutants, deletions, and truncations was done by using Infusion-HD Eco Dry Cloning Kit from Clontech (Mountain View, CA). Wild-type HEG1 cytoplasmic tail (1274-1381), 1364X, C19,

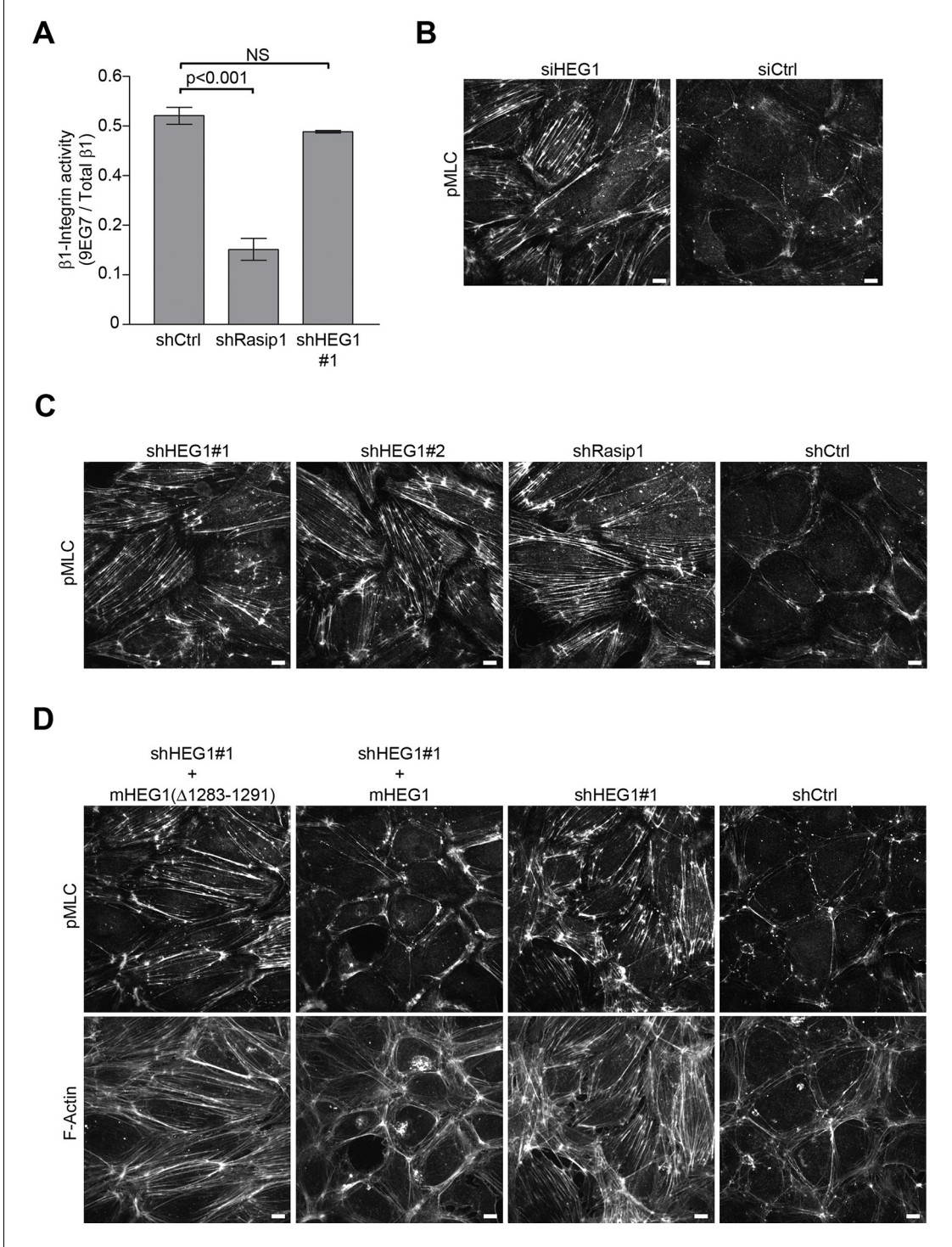

**Figure 6.** Deletion of the Rasip1-binding site in HEG1 prevents suppression of MLC phosphorylation. (**A**) Levels of activated β1 integrin (9EG7) in Rasip1- or HEG1-depleted Human Umbilical Vein Endothelial Cells (HUVEC) were measured by flow cytometry. Depletion of Rasip1 (shRasip1) decreased levels of activated β1 integrin compared to control cells (shCtrl). In contrast, depletion of HEG1 (shHEG1#1) had no effect on levels of activated β1 integrin. Levels of 9EG7 binding were corrected for total β1 integrin expression. Mean values ± SEM of three independent experiments are shown. One-way analysis of variance (ANOVA) with Bonferroni's test was used to compare each condition with control cells (shCtrl). (**B** and **C**) Myosin light chain phosphorylation (pMLC) was analyzed by Spinning Disk Confocal Microscopy (SDCM) in HUVEC, transfected or infected as indicated. siRNA- or shRNA-mediated depletion of HEG1 (B&C; siHEG1, shHEG1#1, shHEG1#2) increased MLC phosphorylation and stress fiber formation. Similarly, lentiviral depletion of Rasip1 (shRasip1) also increased MLC phosphorylation and stress fiber formation. Representative images of 3 independent

*Figure 6 continued on next page*

*Figure 6 continued*

experiments are shown. Scale bars, 10 μm. (**D**) Myosin light chain phosphorylation (pMLC) and F-Actin were analyzed by Spinning Disk Confocal Microscopy (SDCM) in Human Umbilical Vein Endothelial Cells (HUVEC), infected as indicated. Lentiviral depletion of HEG1 (shHEG1) increased levels of pMLC and actin stress fiber formation in HUVEC compared to control cells (shCtrl) which was rescued by expression of FLAG-tagged shHEG1#1-resistant wild-type murine HEG1. In contrast, Rasip1-binding deficient murine HEG1(Δ1283-1291) (corresponding to aa 1327-1335 in human HEG1) failed to rescue pMLC expression and the increase in actin stress fibers. Expression of rescue constructs, analyzed by flow cytometry, is shown in *Figure 6—figure supplement 1*. Representative images of 3 independent experiments are shown. Scale bars, 10 μm.

The following figure supplement is available for figure 6:

**Figure supplement 1.** Expression of murine HEG1 in HUVEC.

and αIIb intracellular tail were previously described (*Gingras et al., 2012*). His6-tagged HEG1 cytoplasmic tail peptides C54, C49, 1334X, 1328X, 1318-1339, and Δ1327-1335 containing an *in vivo* biotinylation peptide tag at the N-terminus were cloned into pET15b. mCherry-HA-Radil was a kind gift from professor S. Angers (University of Toronto, Canada). pcDNA3.1-YFP-ARHGAP29 was a kind gift from professor J.L. Bos (University of Utrecht, The Netherlands). Using pcDNA3.1-YFP-Rasip1 (a kind gift from professor J.L. Bos, University of Utrecht, the Netherlands) as a template for PCR, Rasip1 full-length, 2-265, 266-550, and 551-963 were cloned into p3xFLAG-CMV7.1 or pEGFP-C1 using EcoRI and BamHI restriction sites. Rasip1 Δ334-539 and R182E were generated with two-fragment PCR Infusion cloning using wild-type Rasip1 as a template and cloned into p3xFLAG-CMV7.1 using EcoRI and BamHI restriction sites. Constitutively active Rap1-V12 was cloned into p3xFLAG-CMV7.1 or pEGFP-C1 using EcoRI and BamHI restriction sites Using pcDNA3.1-SNP-FLAG-mHEG1 FL-V5 (modified from a kind gift from professor M.L. Kahn, University of Pennsylvania) as a template, wild-type FLAG-tagged murine HEG1 and Δ1283-1291 (corresponding to aa 1327-1335 in human HEG1) were cloned into pLVX-Het-1 lentiviral expression vector using EcoRI and MluI restriction sites. mHEG1-expressing lentiviral particles were prepared by co-transfection of pLVX-Het-1 mHEG1 WT or Δ1283-1291 with pMDLg/pRRE, pRSV-Rev, and pMD2.G in HEK293T cells. Supernatant, containing lentiviral particles, was collected. Wild-type Rasip1, R182E, and Δ334-539, were cloned into 2K7 lentiviral vector (A kind gift from Alexander Zambon, UC San Diego, USA) using BamHI and XhoI restriction sites. Rasip1-expressing lentiviral particles were prepared by co-transfection of 2K7-Rasip1 wild-type, R182E, or Δ334-539 with pMDLg/pRRE, pRSV-Rev, and pMD2.G in HEK293T cells. Supernatant, containing lentiviral particles, was collected. Mito-mCherry (*Lenter et al., 1993*) and mito-mCherry-HEG1 wild-type or Δ1327-1335 were cloned into pcDNA3.1(-) using XhoI and NheI restriction sites. In vitro transcription to generate mRNA was performed using mMESSAGE mMACHINE® T7 ULTRA Transcription Kit (AM1345, Life Technologies, Grand Island, NY) according to the manufacturer's recommendations.

For lentiviral delivery, HUVEC were grown to 80% confluence on gelatin-coated glass coverslips, and then infected with 2K7-Rasip1-containing lentiviral particles. 72 Hours post-infection, HUVECs were prepared for immunofluorescence analysis (described below). For in vitro binding studies, Rasip1 266-550 was cloned into pETM-41 (EMBL) containing His6 tag and Maltose Binding Protein (MBP) tag using NcoI and BamHI restriction sites.

Oligo's for shRasip1 (Clone TRCN0000437267, target sequence: CCACTGAGTTCTTCCGG-AAAC), shHEG1#1 (Clone TRCN0000253693, target sequence: ACCTTCGTGACAGAGTTTAAA), and shHEG1#2 (Clone TRCN0000253694, target sequence: CATTGGGAGATAGGAGTTATT) are based on the public TRC (The RNAi Consortium, Broad Institute) library and cloned into pLKO.1 using EcoRI and Age1 restriction sites. shRNA-expressing lentiviral particles were prepared by co-transfection of pLKO.1-Rasip1, HEG1#, or HEG1#2 with pMDLg/pRRE, pRSV-Rev, and pMD2.G in HEK293T cells. Supernatant, containing lentiviral particles, was collected. For shRNA delivery, HUVEC were grown to 80% confluence on gelatin-coated glass coverslips, and then infected with shRNA-containing lentiviral particles. 72 Hours post-infection, HUVECs were prepared for immunofluorescence analysis (described below). siRNA specific for HEG1 was from Ambion (Silencer select, ID: s33148). HUVECs were grown to 80% confluence on gelatin-coated glass coverslips, and then transfected with siRNA using Lipofectamine RNAi Max reagent (Life Technologies) according to the

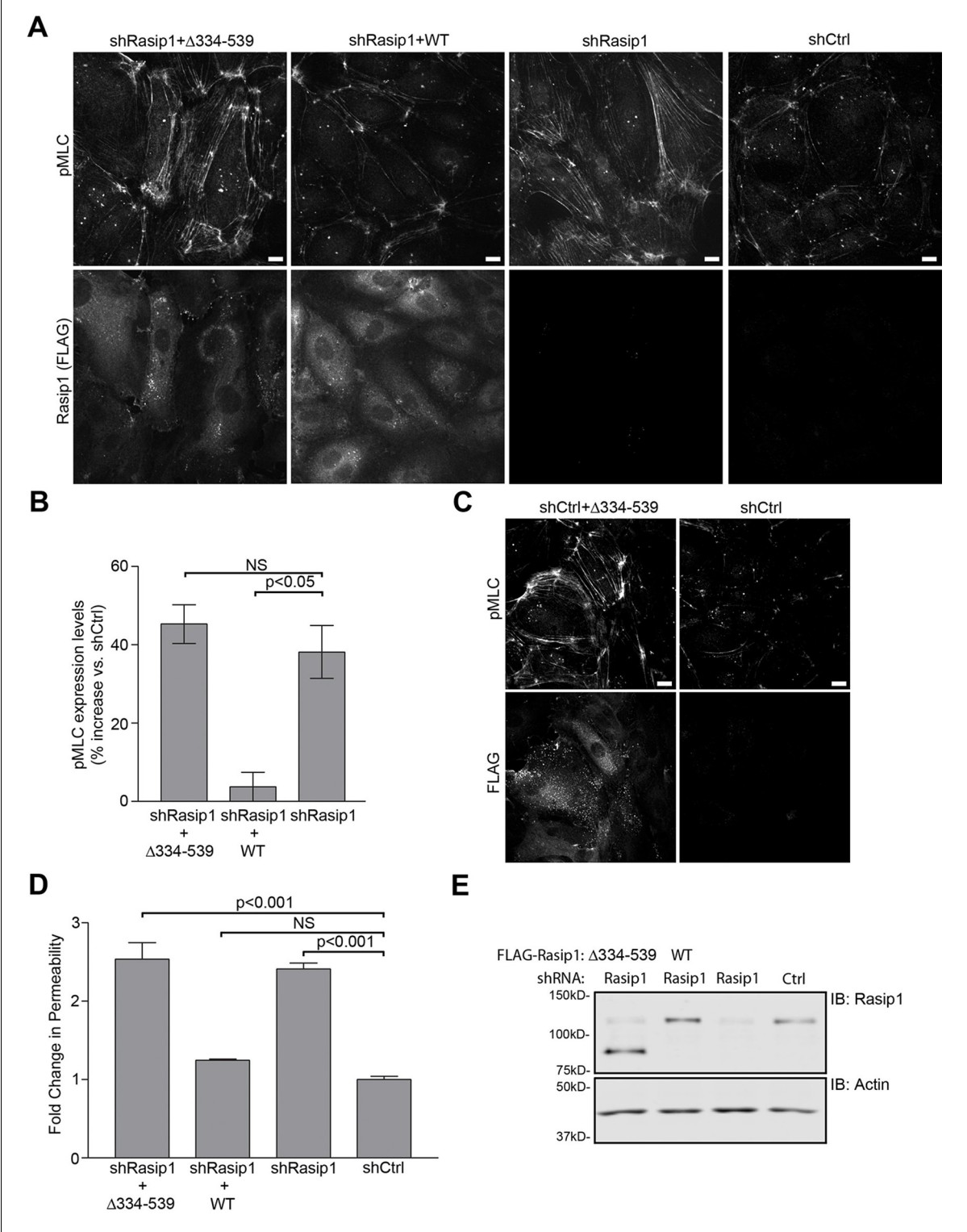

**Figure 7.** HEG1-binding deficient Rasip1 fails to rescue MLC phosphorylation and EC permeability. (**A** and **B**) Myosin light chain phosphorylation (pMLC) was analyzed by Spinning Disk Confocal Microscopy (SDCM) in Human Umbilical Vein Endothelial Cells (HUVEC), infected as indicated (**A**). Integrated Density was measured to quantify levels of pMLC expression (**B**). Lentiviral depletion of Rasip1 (shRasip1) increased levels of pMLC in HUVEC by 40% compared to control cells (shCtrl) which can be rescued by expression of FLAG-tagged shRasip1-resistant wild-type Rasip1. In contrast, HEG1-binding deficient Rasip1(Δ334-539) failed to rescue pMLC expression. Expression of rescue constructs is shown by FLAG staining. Mean values ± SEM are shown. One-way analysis of variance (ANOVA) with Bonferroni's test was used to compare each condition versus Rasip1-depleted cells

*Figure 7 continued on next page*

*Figure 7 continued*

(shRasip1). Data are from 3 independent experiments. Scale bars, 10 µm. (C) pMLC expression was analyzed by SDCM in control HUVEC (shCtrl) or HUVEC expressing FLAG-tagged Rasip1(Δ334-539) (shCtrl+Δ334-539). Expression of Rasip1(Δ334-539) alone induces expression of pMLC similar to Rasip1 knock-down (Panel A). Scale bars, 10 µm. (D and E) Permeability of HUVEC, seeded on fibronectin-coated Transwell filters (pore size 0.4 µm, membrane diam. 12 mm) and infected as indicated, was measured using 70kD-FITC-Dextran (D). Western blot analysis confirmed Rasip1 knock-down and expression of FLAG-tagged rescue constructs (E). Depletion of Rasip1 (shRasip1) increased permeability by two-fold compared to control cells (shCtrl) which can be rescued by expression shRasip1-resitant wild-type Rasip1. In contrast, HEG1-binding deficient Rasip1(Δ334-539) failed to rescue HUVEC permeability. Mean values ± SEM are shown. One-way analysis of variance (ANOVA) with Bonferroni's test was used to compare each condition versus control cells (shCtrl). Data are from 3 independent experiments. See also *Figure 7—figure supplement 1*.

The following figure supplement is available for figure 7:

**Figure supplement 1.** Generating FLAG-Rasip1 shRNA-resistant cDNA.

manufacturer's recommendations. 72 Hours post-transfection, HUVECs were prepared for immuno-fluorescence analysis (described below).

## Cell culture and transfection

HEK293T cells were maintained in Dulbecco's modified Eagle's medium (Corning, Tewksbury, MA) supplemented with 10% FBS (Sigma-Aldrich, Saint Louis, MO), 1% nonessential amino acids, 1% L-Glutamine, and 1% penicillin and streptomycin (all obtained from Invitrogen, Grand Island, NY). Primary Human Umbilical Vein Endothelial Cells (HUVEC) were purchased from Lonza and cultured in EGM2 medium, supplemented with singlequots (Lonza, Walkersville, MD).

HEK293T cells were transfected using *Trans*IT-LT1 transfection reagents according to the manufacturer's recommendations (Mirus Bio, Madison, WI). 24 Hours post-transfection, cells were prepared for Western blot analysis (described below). HUVEC were transfected using TransMessenger Transfection Reagent (301525, QIAGEN, Valencia, CA) according to the manufacturer's recommendations.

## Antibodies and reagents

Polyclonal anti-Rasip1 (Abcam, Cambridge, MA) was used for immunoblotting at 1:1000 and for immunofluorescence at 1:500. Mouse monoclonal (15B2) and rabbit polyclonal anti-KRIT1 (6832) were developed using recombinant KRIT1 FERM domain as the antigen, as previously described (*Glading et al., 2007*). Anti-KRIT1 15B2 was used for immunoprecipitation. Anti-KRIT1 (6832) was used for immunoblotting at 1:1000. Polyclonal-anti-pMLC (Cell Signaling) was used for immunoblotting at 1:1000 and for immune immunofluorescence at 1:300. Monoclonal-anti-VECadherin (Cell Signaling, Danvers, MA) was used for immunofluorescence at 1:1000. Polyclonal or monoclonal anti-FLAG (both from Sigma) were used for immunoblotting at 1:5000 and for immunofluorescence at 1:1000. Monoclonal anti-β-actin (Sigma) was used for immunoblotting at 1:5000. Monoclonal anti-GFP (Clontech) was used for immunoblotting at 1:5000. Monoclonal anti-HA (clone 16B12), monoclonal anti-human CD29 (Clone TS2/16, β1 integrin), and APC-conjugated anti-FLAG (clone L5) were from Biolegend (San Diego, CA). Rat anti-Mouse CD29 (Clone 9EG7, β1 integrin) was purchased from BD Biosciences (5 µg/ml) (San Jose, CA). Secondary AlexaFluor-labelled antibodies for immunofluorescence were from Life Technologies. Secondary AlexaFluor-labelled antibodies for Western blot analysis were from Rockland (Boyerton, PA) or Life Technologies. 8-pCPT-2'-O-Me-cAmp ('007') was purchased from Biolog (Hayward, CA) and used at 100 µM for 20 minutes. Rho Kinase inhibitor H-1152 was purchased from EMD biochemical (Billerica, MA) and used at 3 µM for 30 minutes.

## In vitro protein interaction assay

HEG1 and αIIb intracellular tail proteins were prepared as previously described (*Bear et al., 2000*). In brief, His6-tagged tail proteins, containing an in vivo biotinylation peptide tag at the N-terminus, were expressed and purified from *Escherichia coli*. To perform pull-down assay using cell lysates, Human Umbilical Vein Endothelial Cells (HUVEC) and HEK293T, transfected as indicated, were lysed in cold lysis buffer (50 mM Tris-HCl, pH 7.4, 100 mM NaCl, 10 mM MgCl$_2$, 1%NP40, and 10% glycerol) plus protease and phosphatase inhibitor cocktails (Roche, Indianapolis, IN). A total of 10 µg of

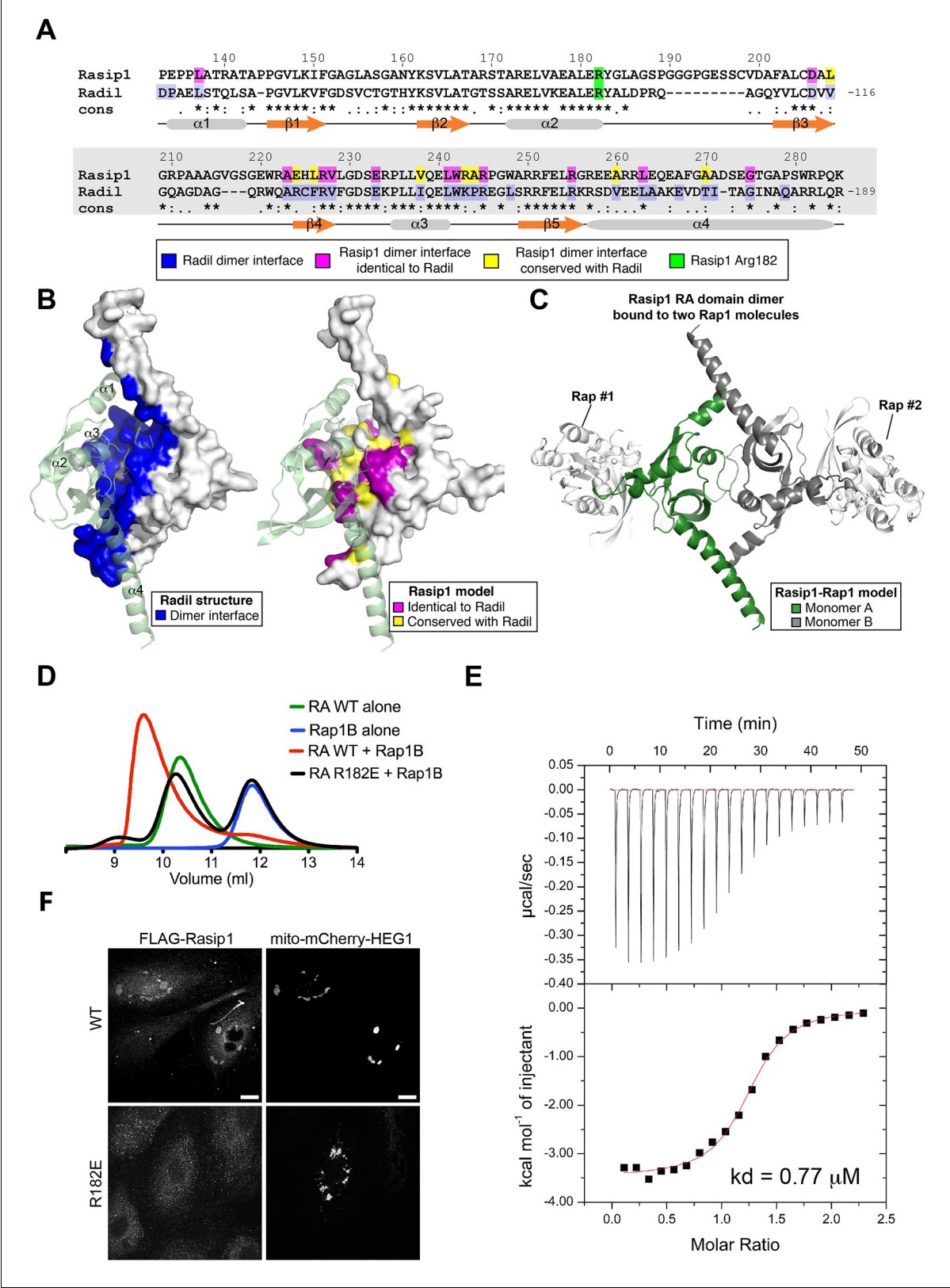

**Figure 8.** Rasip1 RA domain forms a dimer, and Rasip1 Arg182 is important for high affinity Rap1 binding. (**A**) Sequence alignment of human Rasip1 RA domain with human Radil. Symbols denote the degree of conservation: (*) identical, (:) conservative substitution, and (.) semi-conservative substitutions. Secondary structure elements of Radil are shown below the alignment. (**B**) Left: Crystal structure of Radil dimer with the dimer interface highlighted in

*Figure 8 continued on next page*

*Figure 8 continued*

blue (PDB: 3EC8). Right: Residues 134-285 from Rasip1 were modeled using the Radil RA domain crystal structure as a template. View of Rasip1 RA domain dimer with the dimer interface highlighted: identical to Radil (magenta) and conserved (yellow). The residues highlighted are also shown in panel A with the same color code. (C) Model of the Rasip1 dimer with two RA motifs located at opposite ends suggesting it can bind two Rap1 molecules as shown. (D) Binding of the Rasip1 RA domain wild-type and R182E to Rap1 as analyzed on a Superdex-75 (10/300) gel filtration column at room temperature. Incubation of Rasip1 wild-type with Rap1 (red) resulted in complex formation with a shift towards lower volume of elution. In contrast, incubation of Rasip1(R182E) with Rap1 (black) resulted in no interaction with both proteins staying in the free state, suggesting a large reduction in affinity. Furthermore, purified Rasip1 RA domain (50 µM) alone had a large apparent molecular mass (37 kDa; green) as determined by gel filtration compared with a calculated value of 16.3 kDa, suggesting that it forms a dimer in solution. (E) Calorimetric titration of Rap1, out of the syringe into Rasip1 RA domain in the sample cell (kd = 0.77 µM). The titrations were done using monomer concentrations of Rasip1 RA domain. These data show that each Rasip1 dimer can bind two Rap1 monomers. (F) HUVECs, transfected with mito-mCherry-HEG1, were analyzed by Spinning Disk Confocal Microscopy (SDCM) for wild-type (WT) Rasip1 or Rap1-binding deficient Rasip1(R182E) localization which was visualized by FLAG staining. A fraction of wild-type Rasip1, but not Rasip1(R182E), was targeted to mito-mCherry-HEG1 positive structures. Representative images of 3 independent experiments are shown. Scale bars, 10 µm.

immobilized bead-bound proteins was added to 350 µg of clarified lysates. Reactions were kept at 4°C for 4 hours while rotating. After washing the beads with lysis buffer, samples were separated by SDS-PAGE. Bound proteins were detected by immunoblotting.

Human Rasip1, cloned in the pETM-11 (His-tagged) and pETM-41 (His-MBP-tagged) vectors, were expressed in Escherichia coli BL21 Star (DE3) cultured in LB media. Recombinant His-tagged Rasip1 RA domain (residues 134-285), wild-type and R182E polypeptides were purified by nickel-affinity chromatography following standard procedures. The His-tag was removed by cleavage with tobacco etch virus protease overnight, and the protein was further purified by size exclusion chromatography using a Superdex-75. The protein concentration for RA domain monomers was assessed using the A280 extinction coefficient of 24,980 $M^{-1}$. Recombinant His-MBP-fusion Rasip1 central domain (residues 266-550) was purified by nickel-affinity chromatography following standard procedures and dialyzed into PBS. The protein concentration for the His-MBP-Rasip1 central domain was assessed using the A280 extinction coefficient of 26,930 $M^{-1}$. Human Rap1 isoform Rap1B (residues 1–167) cloned into pTAC vector in the Escherichia coli strain CK600K was a generous gift of professor Alfred Wittinghoefer (Max Planck Institute of Molecular Physiology, Germany). The expression, purification and nucleotide exchange for GMP-PNP was achieved as described previously (*Gingras et al., 2013*). His-MBP-Rasip1 central domain at 2.5 µM was incubated with HEG1 matrices in binding buffer (PBS containing 0.1% Triton X-100, 0.5 mM EDTA, 2 mM DTT, and 1 mM Maltose). The mixture was incubated at 6°C for 1 hour on a rotating platform and the beads washed three times with ice cold binding buffer before electroporation. Subsequently, samples were separated by SDS-PAGE and analyzed by Coomassie-blue staining.

## Immunoprecipitation

Human Umbilical Vein Endothelial Cells (HUVEC) were collected in cold lysis buffer (50 mM Tris-HCl, pH 7.4, 100 mM NaCl, 10 mM MgCl$_2$, 1%NP40, and 10% glycerol) plus protease and phosphatase inhibitor cocktails (Roche). A total of 2 µg of monoclonal anti-KRIT1 (15B2) antibody was added to 350 µg of clarified lysates and incubated at 4°C overnight while rotating. Protein G-Sepharose (Invitrogen) was added to the reaction mixture and further incubated for 4 hours at 4°C. After three washes with cold lysis buffer, beads were mixed with sample buffer and proteins were separated by SDS-PAGE. Bound KRIT1 was detected by using polyclonal anti-KRIT1 (6832) antibody.

HEK293T or U2OS cells, transfected and infected as indicated, were collected in cold lysis buffer (50 mM Tris-HCl, pH 7.4, 100 mM NaCl, 10 mM MgCl$_2$, 1%NP40, and 10% glycerol) plus protease and phosphatase inhibitor cocktails (Roche) and subsequently, clarified lysate was incubated with anti-FLAG G1 Affinity Resin (Genscript, Piscataway, NJ; L00432-1) and incubated at 4°C for 3 hours while rotating. After three washes with cold lysis buffer, beads were mixed with sample buffer and proteins were separated by SDS-PAGE.

## Immunofluorescence and calcium switch assay

Human Umbilical Vein Endothelial Cells (HUVEC) were grown to 80% confluence on gelatin-coated glass coverslips, then infected or transfected as indicated. After 72 hours, cells were fixed with 3.7%

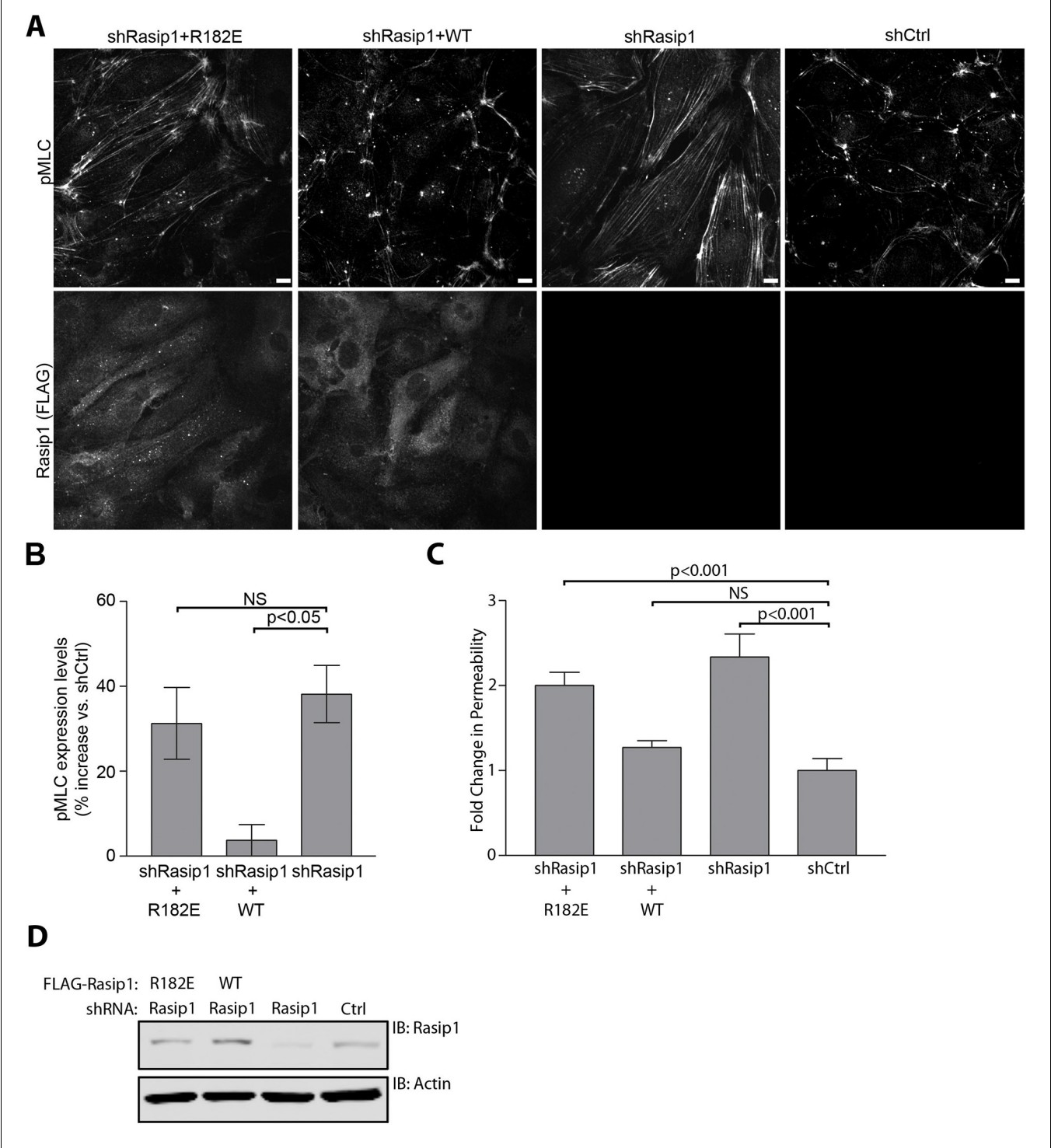

**Figure 9.** Rap1-binding deficient Rasip1 fails to rescue MLC phosphorylation and EC permeability. (**A** and **B**) Myosin light chain phosphorylation (pMLC) was analyzed by Spinning Disk Confocal Microscopy (SDCM) in Human Umbilical Vein Endothelial Cells (HUVEC), infected as indicated (**A**). Integrated Density was measured to quantify levels of pMLC expression (**B**). Depletion of Rasip1 (shRasip1) expression in HUVEC increased levels of pMLC in HUVEC by 40% compared to control cells (shCtrl). This was rescued by expression of FLAG-tagged shRasip1-resistant wild-type Rasip1. In contrast, Rap1-binding deficient Rasip1(R182E) failed to rescue pMLC expression. Expression of rescue constructs is shown by FLAG staining. Mean values ± SEM are shown. One-way analysis of variance (ANOVA) with Bonferroni's test was used to compare each condition versus Rasip1-depleted cells (shRasip1). Data are from 3 independent experiments. Scale bars, 10 μm. (**C** and **D**) Permeability of HUVEC, seeded on fibronectin-coated Transwell filters (pore size 0.4 μm, membrane diam. 12 mm) and infected as indicated, was measured using 70kD-FITC-Dextran (**C**). Western blot analysis confirmed Rasip1

*Figure 9 continued on next page*

*Figure 9 continued*
knock-down and expression of FLAG-tagged rescue constructs (D). Depletion of Rasip1 (shRasip1) increased permeability by two-fold compared to control cells (shCtrl). Permeability was rescued by expression of shRasip1-resistant wild-type Rasip1. In contrast, Rap1-binding deficient Rasip1(R182E) failed to rescue HUVEC permeability. Mean values ± SEM are shown. One-way analysis of variance (ANOVA) with Bonferroni's test was used to compare each condition versus control cells (shCtrl). Data are from 3 independent experiments.

formaldehyde in Phosphate-Buffered-Saline (PBS) for 10 minutes (RT) and subsequently permeabilized with 0.1% Triton X-100 in PBS for 5 minutes (RT). Coverslips were then blocked for 1 hour (37°C) using 10% normal goat serum in PBS. Immunostainings were performed with the indicated antibodies (60 min; RT). Imaging was performed with a Perkin Elmer UltraView Vox Spinning Disk Confocal using a 40x/NA-1.30 or a 60x/NA-1.42 oil objective.

For the calcium switch assay, HUVEC were grown to 80% confluence on gelatin-coated glass coverslips, then infected/transfected as indicated. 72 Hours after infection cells were washed with cold PBS and then incubated with DMEM (5% FBS; 4 mM EGTA) for 30 minutes at 37°C. Cells were then washed with cold PBS and incubated with DMEM (2 mM CaCl; 100 µM 8-pCPT-2'-O-Me-cAmp) for 20 minutes at 37°C. Next, cells were stained as described above. When indicated, to inhibit Rho Kinase activity cells received ROCK inhibitor H-1152 (3 µM) for 30 minutes (37°C) prior to calcium switch assay.

## In vitro endothelial permeability assay

In vitro permeability assay was performed as described previously (*Pfaff et al., 1998*). In brief, membrane filter inserts (pore size 0.4 µm, diameter 12 mm, Corning) were coated with fibronectin (10 µg/ml) for 1 hour at 37°C and placed in a 12-wells plate. Subsequently, fibronectin solution was removed and $1.2 \times 10^5$ Human Umbilical Vein Endothelial Cells (HUVEC) were seeded in each insert. When 80% confluent, cells were infected as indicated and grown for 72 hours. Lower chamber medium was replaced with transport buffer (phenol-free DMEM, 2%FBS). Medium in upper chamber was replaced with transport buffer containing 2 mg/ml 70kD-FITC-Dextran (Sigma). Inserts were then incubated at 37°C and transported sequentially to new plates containing transport buffer at 5 minute intervals for 30 min. FITC in lower chamber medium was measured for each timepoint as well as a 1:1000 dilution of the remaining solution in the upper chamber. Permeability coefficient was calculated for each condition and then normalized as fold change in permeability compared to control HUVEC.

## Mass spectrometry

Lysis Buffer (50 mM Tris-HCl pH 7.4, 100 mM NaCl, 10 mM $MgCl_2$, 1% NP-40, 10% glycerol, 1 mM PMSF and 1x Sigma protease inhibitors) was added at a 1:4 ratio (pellet weight:volume) to a frozen cell pellet (approximately 300mg) of either HeLa or HUVEC cells. The pellet was resuspended by pipetting up and down. 1 µl of benzonase was added per sample and the samples were sonicated for 30secs at amplitude 0.37 (10s pulse, 2s off). The samples were then nutated for 30min at 4°C and subsequently centrifuged at 18K G for 20 min at 4°C. The supernatant was then passed through a 20 µm filter and transferred to a fresh tube. 20 µl of pre-bound streptavidin beads (bound with either HEG1, HEG1ΔYF or ITGA2B) were added to the filtered lysates and incubated for 2 hours at 4°C with gentle agitation. The beads were pelleted by centrifugation (1000g for 1 min), and the supernatant discarded. The beads were washed 1x with 500 µl of cold Lysis Buffer and 2x with 500 µl of cold Wash Buffer (50 mM Hepes-NaOH pH 8.0, 150 mM NaCl, 2 mM EDTA, 0.1% NP40, 10% glycerol). The beads were then transferred to a new tube in 500 µl of 20 mM Tris-HCl, pH 8, and washed again in 500 µl of Tris-HCl. The beads were resuspended in 100 µl of 50 mM ammonium bicarbonate, pH 8.0, containing 500 ng of trypsin, and the mixture was incubated at 37°C with agitation overnight (~15 hours). In the morning, another 500 ng of trypsin was added (in 20 µL of 20 mM Tris-HCl, pH 8), and the sample incubated at 37°C for an additional 4 hours. Finally, the supernatant was collected, and the beads washed twice in 40 µl of water. These washes were pooled with the collected supernatant. Formic acid was added to 2% to stop the digestion. The sample was subsequently vacuumcentrifuged to dryness and resuspended in 5 µl of 5% formic acid for LC/MS-MS. LC-MS/MS was performed using an Eksigent nano Ultra HPLC and a ThermoFisher Orbitrap Velos mass

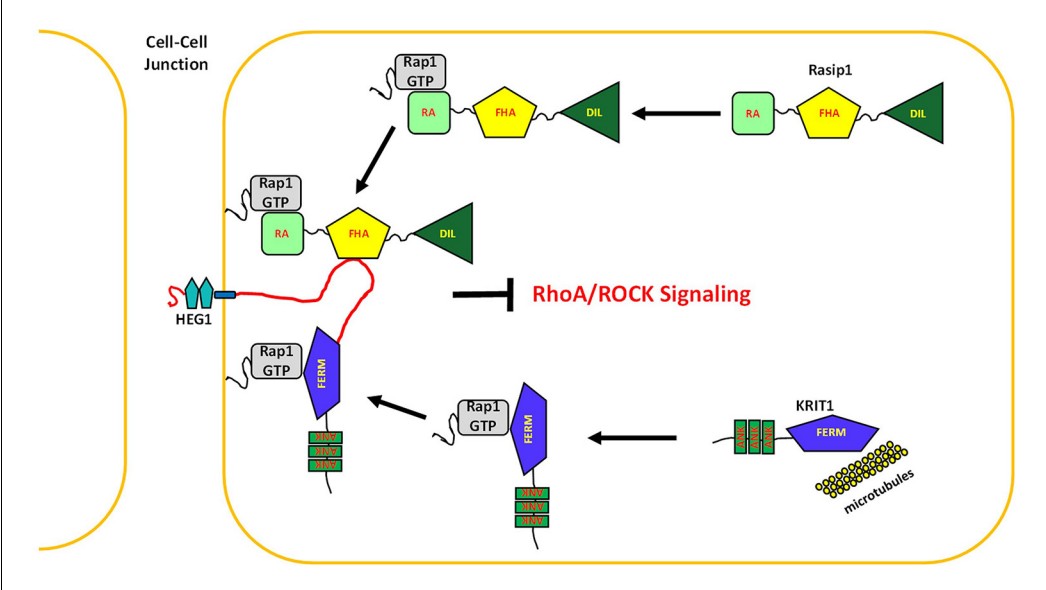

**Figure 10.** HEG1 supports vascular integrity by binding multiple Rap1 effectors. In endothelial cells, Rap1 activation targets Rasip1 and KRIT1 to cell-cell junctions through a direct interaction with the transmembrane receptor HEG1. Here, Rasip1 and KRIT1 regulate junction stability and vascular integrity in part by suppressing ROCK signaling.

spectrometer. Samples were loaded using an autosampler directly onto a home-made 75 µm ID x 10cm packed tip column filled with C18 particles (3 µm, Reprosil, Dr. Maish). Digested peptides were separated over a 90min acetonitrile gradient (2-–35% buffer B over 90 min, 35-80% buffer B over 7 min, hold buffer B at 80% 8 min, and 80-2% B in 2 min) at 200 nl/min. Buffer A was water with 0.1% formic acid; buffer B was 100% ACN with 0.1% formic acid. A data dependent mode was used to acquire an MS scan (60,000 resolution) followed by 10 MS/MS CID scans at low resolution. Peptides were dynamically excluded for 15 sec after being selected for MS/MS. MS/MS spectra were queried against the RefSeq human database (version 53) using Mascot version 2.3.02 (Matrix Science). Methionine residues were searched with a variable modification of +15.9949 Da, and asparagine and glutamine with a variable modification of + 0.984016. Peptides were queried using tryptic cleavage constraints with a maximum of two missed cleavages sites. The mass tolerances were 12 ppm for parent masses and 0.6 Da for fragment masses. Experiments were performed in duplicate and significant interactors were determined using SAINTexpress (*Lopez-Ramirez et al., 2012*). Data was exported to an R-based dotplot tool for visualization (*Teo et al., 2014*).

All RAW mass spectrometry data and downloadable identification and SAINTexpress results tables are deposited in the MassIVE repository housed at the Center for Computational Mass Spectrometry at UCSD (http://proteomics.ucsd.edu/ProteoSAFe/datasets.jsp). The dataset has been assigned the MassIVE ID MSV000079420 and is available for FTP download at: ftp://MSV000079420@massive.ucsd.edu. The dataset was assigned the ProteomeXchange Consortium (http://proteomecentral.proteomexchange.org) identifier PXD003328.

## Size exclusion chromatography

Size exclusion chromatography of recombinant Rasip1 RA domain with Rap1 was performed using a Superdex-75 (10/300) GL (GE Healthcare, Piscataway, NJ) column at room temperature. The proteins, 50 µM each, were mixed in a volume of 100 µl and incubated for 30 min at room temperature before loading onto the column, which was pre-equilibrated with and run with TBS containing 3 mM $MgCl_2$.

## Isothermal Titration Calorimetry

ITC data were collected using a VP-ITC microcalorimeter (MicroCal Ltd., Northampton, MA) at 25°C and analyzed by fitting to a single-site binding equation using MicroCal Origin Software. For Rap1B

binding, 335 µM GMP-PNP-bound Rap1B was titrated from the syringe into the sample cell containing 30µM Rasip1 RA domain. All proteins were dialyzed into 20 mM Sodium Phosphate, 200 mM NaCl, 3 mM MgCl$_2$, pH 6.5 containing 0.1 mM GMP-PNP before performing the experiments.

## Homology model of Rasip1 RA domain

We generated an homology model of the Rasip1 RA domain using the Protein Homology/analogY Recognition Engine (PHYRE V 2.0) (*Antonio Vizcaíno et al., 2015*). The model with the highest confidence score was based on the Radil RA domain structure (PDB 3EC8, 100% confidence, 51% sequence identity). PISA (Proteins, Interfaces, Structures and Assemblies) software suggested that the biological assembly of the Radil RA domain structure is a dimer. The Radil dimer model was generated using the symmetry related molecule from the crystal structure and the Rasip1 RA domain dimer generated using those coordinates. The KRIT1 FERM F1 subdomain in complex with Rap1 (PDB 4HDO) was superimposed with the Rasip1 RA domain (RMSD 1.84 Å2 for 74 amino acids) to generate the Rasip1 RA domain complex with Rap1.

## Flow cytometry

HUVEC were harvested by trypsinization, washed twice with PBS, and subsequently fixed in 2% paraformaldehyde for 10 min at room temperature. After three washes with PBS, cells were transferred into PBS supplemented with 0.5% BSA and 0.5% (wt:vol) saponin) for 20 min. Cells were centrifuged and resuspended in this buffer, containing APC-conjugated anti-FLAG antibody (1:200). After incubation on ice (30 min), cells were washed three times in buffer and then resuspended in PBS for analysis on a BD FACSCalibur (BD Biosciences, San Diego, CA, USA) using CellQuest software (BD Biosciences).

## β1 integrin activation (9EG7 binding)

Cells were harvested by trypsinization, washed twice with PBS, washed once with Tyrode's buffer (125 mM NaCl, 12 mM NaHCO$_3$, 2.5 mM KCl, 1 mM MgCl$_2$ and CaCl$_2$, 5 mM HEPES and Glucose, pH 7.4), then resuspended in Tyrode's buffer containing primary antibodies (9EG7, TS2/16, Mouse IgG, or Rat IgG; 5 µg/ml), and incubated on ice for 30 min. Subsequently, cells were washed three times in Tyrode's buffer, and incubated in Tyrode's buffer containing goat anti-mouse or rat antibodies coupled to Alexa Fluor 647 (1:300) for 30 min on ice. Cells were further incubated with propidium iodide (1 µg/ml), then washed three times in Tyrode's buffer, and analyzed on a BD FACSCalibur (BD Biosciences, San Diego, CA, USA) using CellQuest software (BD Biosciences). Level of 9EG7 (active β1) and TS2/16 (total β1) binding was corrected for background by substracting isotype control values. Results show 9EG7 binding relative to total β1 expression.

## Acknowledgements

We thank Wilma McLaughlin for assistance in molecular biology and Dr AA Bobkov for help with ITC.

## Additional information

### Funding

| Funder | Grant reference number | Author |
|---|---|---|
| National Heart, Lung, and Blood Institute | HL078784,HL 106489, HL 117807, NS 092521 | Mark H Ginsberg |
| Canadian Institutes of Health Research | MOP-123433 | Anne-Claude Gingras |
| American Heart Association | 14POST18180010, 12SDG11610043, 14SDG17870007 | Bart-Jan Dekreuk Alexandre R Gingras Jian J Liu |

The funders had no role in study design, data collection and interpretation, or the decision to submit the work for publication.

## Author contributions

BJdeK, ARG, Conception and design, Acquisition of data, Analysis and interpretation of data, Drafting or revising the article; JDRK, Acquisition of data, Analysis and interpretation of data, Drafting or revising the article; JJL, Conception and design, Analysis and interpretation of data, Contributed unpublished essential data or reagents; ACG, MHG, Conception and design, Analysis and interpretation of data, Drafting or revising the article

## Author ORCIDs

Mark H Ginsberg, http://orcid.org/0000-0002-5685-5417

## Additional files

### Major datasets

The following datasets were generated:

| Author(s) | Year | Dataset title | Dataset URL | Database, license, and accessibility information |
|---|---|---|---|---|
| James DR Knight, Anne-Claude Gingras | 2015 | HEG1 interactions P88 VS3 | http://proteomecentral.proteomexchange.org/cgi/GetDataset?ID=PXD003328 | Publicly available at ProteomeCentral (accession no. PXD003328) |

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
