## [Decision Letter]

Thank you for submitting your work entitled "Heart of Glass Anchors Rasip1 at Endothelial Cell-Cell Junctions to Support Vascular Integrity" for peer review at *eLife*. Your submission has been favorably evaluated by Fiona Watt (Senior editor), a Reviewing editor (Kari Alitalo), and two reviewers.

The reviewers have discussed the reviews with one another and the Reviewing editor has drafted this decision to help you prepare a revised submission.

Summary:

The manuscript reports novel, interesting and important findings that add to our understanding of endothelial cell function. The paper is very clearly written and the study is very systematic and thorough and of a high conceptional and technical level. However, perhaps a greater integration and discussion of the in vitro and in vivo phenotypes associated with loss of HEG, RASIP1 and KRIT1/CCM signaling would provide more insight into the significance of these findings.

Essential revisions:

1) RASIP1-RAP1 function. The authors place a great emphasis on the role of HEG in regulating RASIP1 interaction with RAP1 at EC junctions, but in their last figure they also the converse, i.e. that RAP1 binding is required for RASIP1 to interact with HEG. Can the authors explain further how they believe RASIP1 is interacting with RAP1 and HEG? Do they believe that RASIP1 interacts with RAP1 in a HEG1-independent way and that HEG1's role is simply to localize the proteins spatially? Or does KRIT1 feed RAP1 to RASIP1 to enable interaction with HEG1 even if their binding to HEG is independent per structure-function studies? It would also be valuable to place the RASIP1-ARHGAP29 interaction in context with that between RASIP1 and HEG1 identified by the authors, e.g. does HEG1 interaction alter that between RASIP1 and ARHGAP29?

2) In vivo roles of HEG1, RASIP1 and KRIT1. The in vivo roles of HEG1 and KRIT1 are clearly connected, as zebrafish mutants lacking each protein exhibit identical defects in heart and vascular development and mouse studies show genetic interaction. In contrast, there are no in vivo data that associate the phenotypes of HEG1 and RASIP1 deficient mice. In the final paragraph of their Discussion, the authors suggest that this may be because RASIP1 has functions that are independent of HEG1 but are still required for vascular lumen formation, e.g. integrin activation in ECs. Have the authors tested effects of loss of HEG1-RASIP1 binding on integrin activation and EC adhesion to basement membrane?

3) It is clearly shown that the middle part of Rasip1 (aa 266-550, containing the FHA domain) is required for HEG1 binding, recruitment to cell junctions and for cell contact integrity. However, since the binding area on Rasip is rather large, it would be important to show, that the HEG1 Δ1327-1335 mutant lacking the Rasip1 binding site does indeed affect cell contact integrity. Especially since wt Rasip1 suppresses pMLC not only close to junctions, it would be interesting to see whether it is indeed the junction localization via HEG1 which is necessary for the Rasip1 effects on pMLC and on junction integrity.

---

## [Author Response]

*Essential revisions:*

*1) RASIP1-RAP1 function. The authors place a great emphasis on the role of HEG in regulating RASIP1 interaction with RAP1 at EC junctions, but in their last figure they also the converse, i.e. that RAP1 binding is required for RASIP1 to interact with HEG. Can the authors explain further how they believe RASIP1 is interacting with RAP1 and HEG? Do they believe that RASIP1 interacts with RAP1 in a HEG1-independent way and that HEG1's role is simply to localize the proteins spatially?*

In the initial submission we showed that Rap1 activity promotes recruitment of Rasip1 to cell-cell contacts which is prevented when HEG1 expression is silenced (Figure 3). Furthermore, we show that Rap1-binding is required for Rasip1 recruitment to mitochondrial-targeted HEG1 (Figure 1). In response to this comment, we tested if HEG1 could regulate the interaction between Rap1 and Rasip1. In Figure 3—figure supplement 3, we tested the interaction of Rap1-V12 with Rasip1 in HEG1-silenced cells or by addition of the HEG1 cytoplasmic domain (5uM) peptide and show that neither of these conditions affects the interaction between Rasip1 and Rap1-V12. Thus, HEG1 does not regulate the Rasip1-Rap1 interaction, but does enable Rap1-bound Rasip1 to localize to cell-cell junctions..

*Or does KRIT1 feed RAP1 to RASIP1 to enable interaction with HEG1 even if their binding to HEG is independent per structure-function studies?*

We were unable to discern a role for KRIT1 in regulating the HEG1-Rasip1 interaction *in vitro,* furthermore, in cells, Rasip1 can interact with HEG1 independent of KRIT1 (Figure 2). Silencing KRIT1 did not prevent Rasip1 localization to forming cell-cell contacts or recruitment of Rasip1 to mito-HEG1; (Figure 2 and Figure 3). Thus, it seems unlikely that KRIT1’s role is to feed Rap to Rasip1 to allow its interaction with HEG1. That said, KRIT1 may still contribute to the capacity of Rasip1-HEG1 to inhibit RhoA/ROCK signaling and stabilization of cell-cell junctions.

*It would also be valuable to place the RASIP1-ARHGAP29 interaction in context with that between RASIP1 and HEG1 identified by the authors, e.g. does HEG1 interaction alter that between RASIP1 and ARHGAP29?*

As previous studies have reported that Rasip1 acts by forming a complex with ARHGAP29 and Radil to suppress ROCK signaling and stabilize junctions, we agree that it would be valuable to see whether HEG1 can regulate the interaction between Rasip1 and ARHGAP29 or Radil. We have performed co-immunoprecipitation experiments (Figure 3—figure supplement 3) and tested the binding of Rasip1 with either ARHGAP29 or Radil. As shown in this figure, silencing HEG1 expression does not alter the co-immunoprecipitation of Radil or ARHGAP29 with Rasip1. This indicates that HEG1 is not required for formation of a Rasip1-Radil-ARHGAP29 complex.

*2) In vivo roles of HEG1, RASIP1 and KRIT1. The in vivo roles of HEG1 and KRIT1 are clearly connected, as zebrafish mutants lacking each protein exhibit identical defects in heart and vascular development and mouse studies show genetic interaction. In contrast, there are no in vivo data that associate the phenotypes of HEG1 and RASIP1 deficient mice. In the final paragraph of their Discussion, the authors suggest that this may be because RASIP1 has functions that are independent of HEG1 but are still required for vascular lumen formation, e.g. integrin activation in ECs. Have the authors tested effects of loss of HEG1-RASIP1 binding on integrin activation and EC adhesion to basement membrane?*

*Rasip1* null-mice exhibit vascular collapse which is not evident in the *Heg1* null mice. In the Discussion we suggested that this may be due to Rasip1 functions that are independent of HEG1 such as effects on β1 integrin activation. We have now therefore tested whether HEG1 has effects on binding of 9EG7 antibody as a reporter for β1 integrin activation (Lenter et al., 1993, PNAS). In contrast to Rasip1 silencing, silencing HEG1 expression did not significantly change levels of activated β1 integrin (See new Figure 6).

*3) It is clearly shown that the middle part of Rasip1 (aa 266-550, containing the FHA domain) is required for HEG1 binding, recruitment to cell junctions and for cell contact integrity. However, since the binding area on Rasip is rather large, it would be important to show, that the HEG1 Δ1327-1335 mutant lacking the Rasip1 binding site does indeed affect cell contact integrity. Especially since wt Rasip1 suppresses pMLC not only close to junctions, it would be interesting to see whether it is indeed the junction localization via HEG1 which is necessary for the Rasip1 effects on pMLC and on junction integrity.*

We agree and now report that silencing HEG1 expression has similar effects on ROCK activity and stress fiber formation as Rasip1 silencing. We rescued these phenotypes by expressing shRNA-resistant murine HEG1. In contrast, deletion of the 9 amino acids responsible for Rasip1 binding, rendered HEG1 unable to reverse these phenotypes (See new Figure 6, and Figure 6—figure supplement 1).